# Domain Adaptation meets Individual Fairness. And they get along.

**Debarghya Mukherjee**[*]
Princeton University
University of Michigan
mdeb@umich.edu

**Felix Petersen**[*]
Stanford University
University of Konstanz
mail@felix-petersen.de

**Mikhail Yurochkin**
IBM Research, MIT-IBM Watson AI Lab
mikhail.yurochkin@ibm.com

**Yuekai Sun**
University of Michigan
yuekai@umich.edu

## Abstract

Many instances of algorithmic bias are caused by distributional shifts. For example, machine learning (ML) models often perform worse on demographic groups that are underrepresented in the training data. In this paper, we leverage this connection between algorithmic fairness and distribution shifts to show that algorithmic fairness interventions can help ML models overcome distribution shifts, and that domain adaptation methods (for overcoming distribution shifts) can mitigate algorithmic biases. In particular, we show that (i) enforcing suitable notions of individual fairness (IF) can improve the out-of-distribution accuracy of ML models under the covariate shift assumption and that (ii) it is possible to adapt representation alignment methods for domain adaptation to enforce individual fairness. The former is unexpected because IF interventions were not developed with distribution shifts in mind. The latter is also unexpected because representation alignment is not a common approach in the individual fairness literature.

## 1 Introduction

Although algorithmic bias and distribution shifts are often considered separate problems, there is a recent body of empirical work that shows many instances of algorithmic bias are caused by distribution shifts. Broadly speaking, there are two ways distribution shifts cause algorithmic biases [1]: (i) The model is trained to predict the wrong target; (ii) The model is trained to predict the correct target, but its predictions are inaccurate for demographic groups that are underrepresented in the training data.

From a statistical perspective, the first type of algorithmic bias is caused by *concept* or *posterior drift* between the training data and the real-world. This leads to a mismatch between the model's predictions and actual data. This type of algorithmic bias is also known as *label choice bias* [2]. The second type of algorithmic biases arises when ML models are trained or evaluated with non-diverse data, causing the models to perform poorly on underserved groups. This type of algorithmic bias is caused by a *covariate shift* between the training data and the real-world data. In this paper, we mostly focus on algorithmic biases caused by covariate shift. The overlap between the problems of algorithmic bias and distribution shift suggests two questions:

1. Is it possible to overcome distribution shifts with algorithmic fairness interventions?

2. Is it possible to mitigate biases caused by distribution shifts with domain adaptation methods?

---

[*]Equal Contribution.

36th Conference on Neural Information Processing Systems (NeurIPS 2022).

For a concrete example, consider building an ML model to predict a person's occupation from their biography. For this task, Yurochkin *et al.* [3] showed that ML models trained on top of pre-trained language models without any algorithmic fairness intervention can be unfair: they can change prediction (e.g., from attorney to paralegal or vice versa), when the name and gender pronouns are changed in the input biography. This is a violation of individual fairness (IF), in part caused by underrepresentation of female attorneys in the train (source) data. Consequently, this model underperforms on female attorneys, in particular when female attorneys are better represented in the target domain. This is a type of distribution shift known as subpopulation shift in the domain adaptation literature [4]. In this case, enforcing IF will not only result in a fairer model, but can also improve *performance* in the target domain, i.e., solve the domain adaptation problem.

Now, under the same source and target domains, consider applying a domain adaptation (DA) method that matches the distributions of representations on the domains (see Appendix G for a brief review of DA and algorithmic fairness under distribution shifts). Assuming class marginals are the same[1], i.e., source and target have the same fraction of attorneys, any differences between the source and the target distribution are due to different fractions of male to female attorneys. Learning a feature (representation) extractor that is invariant to gender pronouns and names will align the two domains and result in a model that is individually fair. For group fairness, Schumann *et al.* [5] and Creager *et al.* [6] show that it is possible to leverage DA algorithms to enforce group fairness. The goal of this paper is to complement these results by precisely characterizing the cases in which enforcing IF achieves domain generalization and vice a versa. Our contributions can be summarized as:

1. We show that methods designed for IF can help ML models adapt/generalize to new domains, i.e., improve the accuracy of the trained ML model on out-of-distribution samples.

2. Conversely, we show that DA algorithms that align the feature distributions in the source and target domains can be used to improve IF under certain probabilistic conditions on the features.

We verify our theory on the Bios [7] and the Toxicity [8] datasets: enforcing IF via the methods of Yurochkin *et al.* [3] and Petersen *et al.* [9] improves accuracy on the target domain, and DA methods [10]–[12] trained with appropriate source and target domains improve IF.

## 2   Overcoming Distribution Shift by Enforcing Individual Fairness

The goal of individual fairness is to ensure similar treatment of similar individuals. Dwork *et al.* [13] formalize this notion using $L$-Lipschitz continuity of an ML model $f : \mathcal{X} \to \mathcal{Y}$:

$$d_{\mathcal{Y}}(f(x), f(x')) \leq L d_{\mathcal{X}}(x, x') \tag{2.1}$$

for all $x, x' \in \mathcal{X}$. Here, $d_{\mathcal{Y}}$ is the metric on the output space quantifying the similarity of treatment of individuals, and $d_{\mathcal{X}}$ is the metric on the input space quantifying the similarity of individuals.

Algorithms for enforcing IF are similar to algorithms for domain adaptation/generalization. For example, adversarial training/distributionally robust optimization can not only enforce IF [3], [14], but can also be used for training ML models that are robust to distribution shifts [11], [15]. This similarity is more than a mere coincidence: the goal in both enforcing IF and domain adaptation/generalization is *ignoring uninformative dissimilarity*. In IF, we wish to ignore variation among inputs that are attributed to variation of the sensitive attribute. In domain adaptation/generalization, we wish to ignore variation among inputs that are attributed to the idiosyncracies of the domains. Mathematically, ignoring uninformative dissimilarity is enforcing invariance/smoothness of the ML model among inputs that are dissimilar in uninformative ways. For example, (2.1) requires the model to be approximately constant on small $d_{\mathcal{X}}$-balls.

In this section, we exploit this connection between IF and domain adaptation/generalization to show that enforcing IF can improve accuracy in target domain under covariate shift *if the regression function is individually fair*. In order words, if the inductive bias from enforcing IF is correct, then enforcing IF improves accuracy in the target domain. More concretely, we consider the task of adapting an ML model from a source domain to a target domain. We have $n_s$ labeled samples from the source domain $\{(x_{s,i}, y_{s,i})\}_{i=1}^{n_s}$ and $n_t$ unlabeled samples from the target domain $\{x_{t,i}\}_{i=1}^{n_t}$. Our goal is to

---

[1]This setting corresponds to a domain shift assumption common in the DA literature.

obtain a model $\widehat{f} \in \mathcal{F}$ that has comparable accuracy on the source and target domains. We assume the **regression function** $f_0(x) \triangleq \mathbf{E}\big[y_i \mid x_i = x\big]$ in the source and target domains are identical.

$$y_{e,i} = f_0(x_{e,i}) + \epsilon_{e,i}, \quad e \in \{s, t\}, \tag{2.2}$$

where $\epsilon_i$'s are exogenous error terms with mean zero and variance $\sigma_e^2$. This is a special case of distribution shift called **covariate shift** [16]. The covariate shift problem is most challenging when the model class is *mis-specified* (i.e., $f_0 \notin \mathcal{F}$) and this is the primary focus of this paper. As an example, consider the Inclusive Images Challenge [17]. Publicly available image datasets often lack geo-diversity. Thus, ML models trained on such datasets tend to make mistakes on images from underrepresented countries. As a concrete example, while brides in western countries typically wear white dresses at wedding ceremonies, brides in non-western countries may not. An ML model trained on images from mostly western countries may not recognize brides from other parts of the world that are not wearing white dresses. Although there is a function (on images) that recognizes brides from non-western countries (e.g., the function humans implicitly use to recognize brides), the ML model does not learn this function because either the function is not in the model class and/or the inductive bias of the learning algorithm leads the algorithm to pick a different function (i.e., inductive bias of learning algorithm is mis-specified).

To warm up, we consider the transductive (learning) setting before moving on to the inductive setting. Recall that, in the transductive setting, the learner is given a set of labeled samples and another set of unlabeled samples. The goal is correctly predicting the labels of the given unlabeled samples; the learner is unconcerned with the accuracy of the model on new test samples. This is different from the inductive setting, where the goal is correctly predicting the labels of new test samples. The features of the unlabeled samples (but not their labels) are used for training in both settings. We provide theoretical results for both settings.

## 2.1 Warm Up: The Transductive Setting

In the transductive setting, we are only concerned with the accuracy of the predictions on the unlabeled samples from the target domain in the training data. The distribution of unlabeled samples is different from the (marginal) distribution of features in the source domain due to covariate shift. Thus, the problem is similar to that of extrapolation/label propagation in which we wish to propagate the labels/signal from the labeled samples in the source domain to the unlabeled samples in the target domain. Towards this goal, we leverage the (labeled) source and (unlabeled) target samples and the inductive bias on the smoothness of the regression function. We encode this inductive bias in a regularizer $\mathcal{R}$ and solve the following regularized risk minimization problem

$$\hat{f} = \arg\min_{f \in \mathcal{F}} \left[ \frac{1}{n_s} \sum_{i=1}^{n_s} \mathcal{L}\left(y_i, f(x_i)\right) + \lambda \mathcal{R}_n\left(f(X)\right) \right] \tag{2.3}$$

where $\mathcal{F}$ is the model class, $\mathcal{L}$ is a loss function, and $\lambda > 0$ is a regularization parameter. In the transductive setting, the regularizer $\mathcal{R}_n$ is a function of the vector of model outputs on the source and target inputs: $f(X) \triangleq \big[f(X_s)^\top, f(X_t)^\top\big]^\top$, where $f(X_s) \in \mathbf{R}^{n_s}$ (resp. $f(X_t) \in \mathbf{R}^{n_t}$) is the vector of outputs on the source (resp. target) inputs. Intuitively, the regularizer enforces invariance/smoothness of the model outputs on the source and target inputs.

A concrete example of a such a regularizer is the **graph Laplacian regularizer**. A graph Laplacian regularizer is based on a similarity symmetric kernel $K$ on the input space $\mathcal{X}$. For example, Petersen *et al.* [9] take kernel $K$ to be a decreasing function of a fair metric that is learned from data [18], e.g., a metric in which the distance between male and female biographies with similar relevant content is small. In domain adaptation, a similar intuition can be applied. For example, suppose the source train data consists of Poodle dogs and Persian cats (the task is to distinguish cats and dogs), and the target data consists of Dalmatians and Siamese cats [19]. Then, a meaningful metric for constructing kernel $K$ assigns small distances to different breeds of the same species.

Given the kernel, we construct the similarity matrix $\mathbf{K} = \left[K\left(X_i, X_j\right)\right]_{i,j=1}^{n}$. Note that, here, we are considering all the source and target covariates together. Based on the similarity matrix, the (unnormalized) Laplacian matrix is defined as $\mathbf{L} = \mathbf{D} - \mathbf{K}$ where $\mathbf{D}$ is a diagonal matrix with $\mathbf{D}_{i,i} = \sum_j K(X_i, X_j)$, which is often denoted as the degree of the $i^{th}$ observation. There are also other ways of defining $\mathbf{L}$ (e.g., $\mathbf{L} = \mathbf{D}^{-1/2}\mathbf{K}\mathbf{D}^{-1/2}$ or $\mathbf{L} = \mathbf{I} - \mathbf{D}^{-1}\mathbf{K}$) which would also lead to

the similar conclusion, but we stick to unnormalized Laplacian for the ease of exposition. Based on the Laplacian matrix $\mathbf{L}$, we define the graph Laplacian regularizer $\mathcal{R}$ as:

$$\mathcal{R}_n(f(X)) = \tfrac{1}{n^2} f(X)^\top \mathbf{L} f(X) = \tfrac{1}{n^2} \sum_{i,j} K(X_i, X_j) \left(f(X_i) - f(X_j)\right)^2.$$

The above regularizer enforces that if $K(X_i, X_j)$ is large for a pair $(X_i, X_j)$ (i.e., they are similar), $f(X_i)$ must be close to $f(X_j)$. As mentioned earlier, for individual fairness, $K(X_i, X_j)$ is chosen to be a monotonically decreasing function of $d_{\text{fair}}(X_i, X_j)$, which ensures that $f(X_i)$ and $f(X_j)$ are close to each other when $X_i$ is close to $X_j$ with respect to the fair metric (for more details, see Petersen *et al.* [9]). Recently, Lahoti *et al.* [20], Kang *et al.* [21], and Petersen *et al.* [9] used the graph Laplacian regularizer to post-process ML models so that they are individually fair. This is also widely used in semi-supervised learning to leverage unlabeled samples [22].

We focus on problems in which the model class $\mathcal{F}$ is mis-specified, i.e., $f_0 \notin \mathcal{F}$. If the model is well-specified (i.e., $f_0 \in \mathcal{F}$), the optimal prediction rule in the training and target domains are identical (both are $f_0$). It is possible to learn the optimal prediction rule for the target domain from the training domain (e.g., by empirical risk minimization (ERM)), and there is no need to adapt models trained in the source domain to the target. On the other hand, if the model is mis-specified, the transfer learning task is non-trivial because the optimal prediction rule model depends on the distribution of the inputs (which differ in training and target domains). Here, we focus on the non-trivial case. We show that, as long as $f_0$ satisfies the smoothness structure enforced by the regularizer, $\widehat{f}$ from (2.3) remains accurate at the target inputs $\{x_{t,i}\}_{i=1}^{n_t}$. First, we state our assumptions on the loss function $\mathcal{L}$ and the regularizer $\mathcal{R}_n$.

**Assumption 2.1.** *We assume that the regression function is smooth with respect to the penalty $\mathcal{R}_n$, i.e., $\mathcal{R}_n(f_0(X)) \leq \delta$ for some small $\delta > 0$.*

This is an assumption on the effect of the smoothness structure enforced by the regularizer being in agreement with the regression function $f_0$.

**Assumption 2.2.** *We assume that $\mathcal{R}$ is $\frac{\mu_{\mathcal{R}_n}}{n_t}$-strongly convex with respect to the model outputs on the target inputs and $\frac{L_{\mathcal{R}_n}}{n}$-strongly smooth. More specifically, for $v_1 \in \mathbf{R}^{n_s}, v_2, v \in \mathbf{R}^{n_t}, \tilde{v}, v_0 \in \mathbf{R}^n$*

$$\mathcal{R}_n\left(v_1, v_2\right) \geq \mathcal{R}_n\left(v_1, v\right) + \langle v_2 - v, \partial_t \mathcal{R}_n\left(v_1, v\right)\rangle + \frac{\mu_{\mathcal{R}_n}}{2n_t} \|v_2 - v\|_2^2.$$

$$\mathcal{R}_n\left(v_1, v_2\right) \leq \mathcal{R}_n\left(v, \tilde{v}\right) + \left\langle \begin{pmatrix} v_1 - v \\ v_2 - \tilde{v} \end{pmatrix}, \partial \mathcal{R}_n\left(v, \tilde{v}\right)\right\rangle + \frac{L_{\mathcal{R}_n}}{2n} \left\| \begin{bmatrix} v_1 - v \\ v_2 - \tilde{v} \end{bmatrix} \right\|_2^2.$$

This is a regularity assumption on the regularizer to ensure the **extrapolation map** $y_t : \mathbf{R}^{n_s} \to \mathbf{R}^{n_t}$

$$y_t^*(v) \triangleq \arg\min_{t \in \mathbf{R}^{n_t}} \mathcal{R}_n(v, t) \tag{2.4}$$

is well-behaved. Intuitively, the extrapolation map extrapolates (hence its name) model outputs on the source domain to the target domain *in the smoothest possible way*. Next, we state our assumptions on the loss function:

**Assumption 2.3.** *The loss function $\mathcal{L} : \mathbf{R} \times \mathbf{R} \to \mathbf{R}_+$ satisfies $\mathcal{L}(a, b) \geq 0$ and $= 0$ if and only if $a = b$. Furthermore, it is $\mu_{\mathcal{L}}$ - strongly convex and $L_{\mathcal{L}}$ - strongly smooth, i.e.,*

$$\mathcal{L}(x, y) \geq \mathcal{L}(x_0, y_0) + \langle (x, y) - (x_0, y_0), \partial \mathcal{L}(x_0, y_0)\rangle + \tfrac{\mu_{\mathcal{L}}}{2} \|(x, y) - (x_0, y_0)\|_2^2.$$

$$\mathcal{L}(x, y) \leq \mathcal{L}(x_0, y_0) + \langle (x, y) - (x_0, y_0), \partial \mathcal{L}(x_0, y_0)\rangle + \tfrac{L_{\mathcal{L}}}{2} \|(x, y) - (x_0, y_0)\|_2^2.$$

Assumption 2.3 is standard in learning theory, which provides us control over the curvature of the loss function.

**Theorem 2.4.** *Suppose $\hat{f}$ is the estimated function obtained from (2.3). Under Assumption 2.3 on the loss function and Assumptions 2.1 and 2.2 on the regularizer, we have the following bound on the risk in the target domain:*

$$\tfrac{1}{n_t} \sum_{i=1}^{n_t} \mathcal{L}\left(\hat{f}(x_{t,i}), f_0(x_{t,i})\right) \leq \alpha_n\Big[\tfrac{1}{n_s}\sum_{i=1}^{n_s} \mathcal{L}\left(\hat{f}(x_{s,i}), f_0(x_{s,i})\right) + \lambda \mathcal{R}_n(\hat{f}(X))\Big] + \beta_n \mathcal{R}_n(f_0(X)). \tag{2.5}$$

*where*

$$\alpha_n = \max\left\{ \frac{L_{\mathcal{L}} L_{\mathcal{R}}^2 \left(\mu_{\mathcal{L}} + 3L_{\mathcal{L}}\right)}{2\mu_{\mathcal{R}}^2 \mu_{\mathcal{L}}} \rho_n, \; \frac{2 + L_{\mathcal{L}}}{\lambda \mu_{\mathcal{R}}}(1 + \rho_n) \right\}, \quad \beta_n = \frac{2 + L_{\mathcal{L}} + L_{\mathcal{L}}^2}{\mu_{\mathcal{R}}}(1 + \rho_n). \tag{2.6}$$

*with $\rho_n = n_s/n_t$.*

We note that the right side of (2.5) does *not* depend on the $y_{i,s}$'s in the target domain. Intuitively, Theorem 2.4 guarantees the accuracy of $\widehat{f}$ on the inputs from the target domain as long as the following conditions hold.

1. The model class $\mathcal{F}$ is rich enough to include an $f$ that is not only accurate on the training domain, but also satisfies the smoothness/invariance conditions enforced by the regularizer. This implies the first term on the right side of (2.5) is small.

2. The exact relation between inputs and outputs encoded in $f_0$ satisfies the smoothness structure enforced by the regularizer. This implies the second term on the right side of (2.5) is small.

If the model is correctly specified ($f_0 \in \mathcal{F}$) and the regression function perfectly satisfies the smoothness conditions enforced by the regularizer ($\mathcal{R}_n(f_0) = 0$), then the bias term vanishes. In other words, Theorem 2.4 is *adaptive* to correctly specified model classes.

**Example: Laplacian regularizer** We now show that the graph Laplacian regularizer satisfies Assumption 2.2. As $\mathcal{R}_n(f(X))$ is a quadratic function of $\mathbf{L}$, it is immediate that $n\nabla^2 \mathcal{R}_n(f(X)) = \mathbf{L}$. Therefore, the strong convexity and smoothness of $\mathcal{R}$ depend on the behavior of the maximum and minimum eigenvalues of $\mathbf{L}$. The maximum eigenvalue of $\mathbf{L}$ is bounded above for the fixed design, which plays the role of $L_{\mathcal{L}}/2$ in Assumption 2.2. For the lower bound, we note that we only assume strong convexity with respect to the target samples fixing the source samples. If we divide the whole Laplacian matrix into four blocks, then the value of the regularizer in terms of these blocks will be:

$$\mathcal{R}_n(f(X)) = \sum_{i,j \in \{s,t\}} f(X_i)^\top \mathbf{L}_{ij} f(X_j) \,.$$

Therefore, the Hessian of $\mathcal{R}_n$ with respect to the model outputs in the target domain is $\mathbf{L}_{TT}$ whose minimum eigenvalue is bounded away from 0 as long as the graph is connected, i.e., source inputs have a degree of similarity with target inputs. Thus, $\mathcal{R}_n$ satisfies Assumption 2.2. Graph Laplacian regularizer is often used to achieve individual fairness [9], [20], [21] and our Theorem 2.4 shows that it can also be used for domain adaptation. We further verify this empirically in Section 2.4.

*Proof Sketch of Theorem 2.4.* To keep things simple, we focus on the case in which the loss function is quadratic ($\mathcal{L}(x,y) = \frac{1}{2}(x-y)^2$). We have

$$\frac{1}{2n_t}\|\widehat{f}(X_t) - f_0(X_t)\|_2^2 \lesssim \frac{1}{2n_t}\|\widehat{f}(X_t) - y_t^*(\widehat{f}(X_s))\|_2^2 + \frac{1}{2n_t}\|y_t^*(\widehat{f}(X_s)) - y_t^*(f_0(X_s))\|_2^2$$
$$+ \frac{1}{2n_t}\|y_t^*(f_0(X_s)) - f_0(X_t)\|_2^2. \quad (2.7)$$

The first term depends on the smoothness of the model outputs across the source and target domain $\widehat{f}(X)$: it measures the discrepancy between the model outputs in the target domain $f(X_t)$ and the smoothest extrapolation of the model outputs in the source domain to the target domain $y_t^*(f(X_s))$. Similarly, the third term depends on the smoothness of the regression function (across the source and target domains). In Appendix B.1, we bound the two terms with $\mathcal{R}(\widehat{f}(X))$ and $\mathcal{R}(f_0(X))$.

It remains to bound the second term in (2.7). Intuitively, stability of the extrapolation map (2.4) implies the extrapolation operation is similar to a projection onto smooth functions, so the second term satisfies $\frac{1}{2n_t}\|y_t^*(\widehat{f}(X_s)) - y_t^*(f_0(X_s))\|_2^2 \lesssim \frac{1}{2n_s}\|\widehat{f}(X_s) - f_0(X_s)\|_2^2$. See Appendix B.1. $\square$

## 2.2 The Inductive Setting

We now consider the inductive setting. Previously, in Section 2.1, we focused on the accuracy of the fitted model $\widehat{f}$ on the inputs from the test domain $\{x_{t,i}\}_{i=1}^{n_t}$. Here we instead consider the *expected* loss of $\widehat{f}$ at a new (previously unseen) input point in the target domain. We consider a problem setup similar to that in Section 2.1: the $n_s$ labeled samples from the source domain are independently drawn from the source distribution $P$, while the $n_t$ unlabeled samples from the target domain are independently drawn from (the marginal of) the target distribution $Q$. We also assume the covariate shift condition (2.2). The method remains the same as before: we learn $\widehat{f}$ from (2.3).

The main difference between the inductive and transductive settings is in the population version of the regularizer: In the transductive setting, we are only concerned with the output of the ML model for the inputs in the source and target domains; thereby, the population version of the regularizer remains a function of (the vector of) model outputs on the inputs in the source and target domains. In the

inductive setting, we are also concerned with the output of the ML model on previously unseen points; thus, we consider the regularizer as a *functional* (i.e., a higher order function): $\mathcal{R} : \mathcal{F} \times \mathcal{F} \to \mathbf{R}$ (the two arguments corresponds to $f(X_s)$ and $f(X_t)$ in the transductive case). For example, the population version of the graph Laplacian regularizer (in the inductive setting) is

$$\mathcal{R}(f, g) \triangleq \mathbf{E}\left[\tfrac{1}{2}(f(X_s) - g(X_t))^2 K(X_s, X_t)\right],$$

where $X_s \sim P_X$ and $X_t \sim Q_X$. The population version of (2.3) in the inductive setting is

$$\tilde{f} \triangleq \arg\min_{f \in \mathcal{F}} \mathbf{E}[\mathcal{L}(Y_s, f(X_s))] + \lambda \mathcal{R}(f, f). \tag{2.8}$$

Now we state the assumptions to extend Theorem 2.4 to the inductive setting.

**Assumption 2.5.** *The function $f_0$ satisfies $\mathcal{R}(f_0, f_0) \leq \delta$ for some small $\delta > 0$.*

**Assumption 2.6.** *The (population) regularizer $\mathcal{R}$ satisfies the following strong convexity condition:*

$$\mathcal{R}(f, g_1) \geq \mathcal{R}(f, g_2) + \partial_2 \mathcal{R}\left((f, g_2); g_1 - g_2\right) + \frac{\mu_\mathcal{R}}{2}\|g_1 - g_2\|_Q^2,$$

*and the following Lipschitz condition on the partial derivative of $\mathcal{R}$ with respect to the second coordinate, i.e., for any two $f_1, f_2$:*

$$|\partial_2 \mathcal{R}((f_1, g); h) - \partial_2 \mathcal{R}((f_2, g); h)| \leq \mathcal{L}_\mathcal{R}\|f_1 - f_2\|_P\|h\|_Q,$$

*for some constants $\mu_\mathcal{R}, \mathcal{L}_\mathcal{R} > 0$. Here, $\partial_2 \mathcal{R}((f, g); h)$ indicates the Gateaux derivative of $\mathcal{R}$ with respect to the second coordinate along the direction $h$.*

Assumptions 2.5 and 2.6 are analogues of Assumptions 2.1 and 2.2 in the inductive setting. In fact, it is possible to show that Assumptions 2.5 and 2.6 imply Assumptions 2.1 and 2.2 with high probability by appealing to (uniform) laws of large numbers (see Appendix D). The following theorem provides a bound on the population estimation error of $\tilde{f}$ on the target domain:

**Theorem 2.7.** *Under Assumptions 2.3, 2.5, and 2.6, we have:*

$$\mathbb{E}_Q[\mathcal{L}(\widetilde{f}(x), f_0(x)] \leq C_1 \left[\mathbb{E}_P[\mathcal{L}(\widetilde{f}(x), f_0(x)] + \lambda \mathcal{R}(\tilde{f}, \tilde{f})\right] + C_2 \mathcal{R}(f_0, f_0).$$

*for some constants $C_1, C_2$ defined in the proof.*

The bound obtained in Theorem 2.7 is comparable to (2.5): the right side does not depend on the distribution $Y_Q \mid X_Q$. The second term denotes the aptness of regularizer $\mathcal{R}$, i.e., how well it captures the smoothness of $f_0$ over the domains. Similar to (2.5), we note that the bound in Theorem 2.7 is adaptive to correctly specified model classes.

To wrap up, we compare our theoretical results to other theoretical results on domain adaptation. There is a long line of work started by Ben-David *et al.* [23] on out-of-distribution accuracy of ML models [10], [24]–[27]. Such bounds are usually of the form

$$\mathbb{E}_Q\left[\mathcal{L}(f(x), f_0(x))\right] \lesssim \mathbb{E}_P\left[\mathcal{L}(f(x), f_0(x))\right] + \mathrm{disc}(P, Q) \tag{2.9}$$

for any $f \in \mathcal{F}$, where $\mathrm{disc}(P, Q)$ is a measure of discrepancy between the source and target domains. For example, Zhang *et al.* [26] show (2.9) with

$$\mathrm{disc}(P, Q) \triangleq \sup_{f, f' \in \mathcal{F}} \left\{\mathbf{E}_Q\left[\mathcal{L}(f(X), f'(X)\right] - \mathbf{E}_P\left[\mathcal{L}(f(X), f'(X)\right] \right\}.$$

A key feature of these bounds is that it is possible to evaluate the right side of the bounds with unlabeled samples from the target domain (and labeled samples from the source domain). Compared to our bounds, there are two main differences:

1. Equation 2.9 applies to any $f \in \mathcal{F}$ (while our bound only applies to a specific $\widetilde{f}$ from (2.8)). Although this uniform applicability is practically desirable (because it allows practitioners to evaluate the bound *a posteriori* to estimate the out-of-distribution accuracy of the trained model), it precludes the bounds from adapting to correct specification of the model class.

2. The uniform applicability of the bound (to any $f \in \mathcal{F}$) also precludes (2.9) from capturing the effects of the regularizer.

**Remark 2.8.** *Although our theoretical analysis in the main paper is under the assumption of covariate shift, our results can certainly be extended to the case when the conditional mean function $\mathbb{E}[Y \mid X]$ is different on different domains. We present an extension of Theorem 2.7 to this effect in Appendix E. Other theorems (e.g., Theorem 2.4) can also be extended using analogous arguments.*

## 2.3 Extension to Domain Generalization

In this subsection, we further extend our results to the domain generalization setup, i.e., when we have no observations from the target domain. In the previous domain adaptation setup, when we had access to unlabeled data from the target domain, we used a suitable regularizer to extrapolate the prediction performance from the source domain to the target domain. However, when we do not have unlabeled data from the target domain, we need to alter the regularizer appropriately, so that we have some uniform guarantee over all domains in the vicinity of the source domain. Here is an example of a regularizer that seeks to improve domain generalization:

$$\mathcal{R}(f,g) = \left\{ \max_T \quad \mathbb{E}_{X \sim P} \left[ (f(X) - g(T(X)))^2 \right] \qquad \text{s.t.} \quad \mathbb{E}_{X \sim P} \left[ \|X - T(X)\| \right] \leq \epsilon. \quad (2.10) \right.$$

$T$ here can be thought as an adversarial map that maps $X$ to an adversarial example $X' = T(X)$ that maximizes the difference $f(X) - g(X')$. As we need some uniform guarantees across all domains in the vicinity of the source domain, $T$ produces the adversarial test domain example. This regularizer is similar to the SenSeI regularizer originally proposed and studied by Yurochkin *et al.* [3] for enforcing individual fairness. In fact, $\mathcal{R}(f, f)$ is exactly the (Mongé form) of the SenSeI regularizer. Note that we can further generalize this regularizer by incorporating a general loss function $\mathcal{L}$ in the first equation or a general metric $d$ in the second equation. However, as this does not add anything to the underlying intuition, we confine ourselves to the $\ell_2$ metric here. Next, we present our theoretical findings with respect to this regularizer. To this end, we define the set of transformations $\mathcal{T}_\epsilon = \{T : \mathbb{E}_{x \sim P} [\|x - T(x)\|] \leq \epsilon\}$ and the corresponding set of measures $\mathcal{Q}_\epsilon = \{Q : T \# P = Q, T \in \mathcal{T}_\epsilon\}$. We show that it is possible to generalize the performance of the estimator $\hat{f}$ obtained in (2.3) uniformly over the measures in $\mathcal{Q}_\epsilon$. As mentioned previously, we only work with the quadratic loss function, but our result can be extended to the general loss function. The following theorem establishes a uniform bound on the estimation error of the population function $\tilde{f}$ obtained from (2.3) with the regularizer as defined in (2.10):

**Theorem 2.9.** *The population estimator $\tilde{f}$ satisfies the following bound on the estimation error:*

$$\sup_{Q \in \mathcal{Q}_\epsilon} \mathbb{E}_{x \sim Q} \left( \tilde{f}(x) - f_0(x) \right)^2 \leq 4 \left[ R(\tilde{f}, \tilde{f}) + R(f_0, f_0) + \mathbb{E}_{x \sim P} \left( \tilde{f}(x) - f_0(x) \right)^2 \right].$$

The bound obtained in the above is the same as the one obtained in Theorem 2.7 (up to constants) and has analogous interpretation: it consists of the minimum training error achieved on $\mathcal{F}$ and the smoothness of $f_0$ quantified in terms of the regularizer. Moreover, the bound holds uniformly over all the domains $Q \in \mathcal{Q}_\epsilon$, i.e., the performance of the estimator $\hat{f}$ can be extrapolated to all the domains in $\mathcal{Q}_\epsilon$, provided that $\mathcal{R}(f_0, f_0)$ is small.

## 2.4 Empirical Results

We verify our theoretical findings empirically. Our goal is to improve performance under distribution shifts using individual fairness methods. We consider SenSeI [3], Sensitive Subspace Robustness (SenSR) [14], Counterfactual Logit Pairing (CLP) [28], and GLIF [9]. GLIF, similar to domain adaptation methods, requires unlabeled samples from the target. The other methods only utilize the source data as in the domain generalization scenario. Our theory establishes guarantees on the target domain performance for SenSeI (Section 2.3) and GLIF (Section 2.1).

**Datasets and Metrics** We experiment with two textual datasets, Toxicity [8] and Bios [7]. In Toxicity, the goal is to identify toxic comments. This dataset has been considered by both the domain generalization community [4], [6], [29] (under the name Civil Comments) as well as the individual fairness community [3], [9], [28]. The key difference between the two communities are in the comparison metrics. In domain generalization, it is common to consider performance on underrepresented groups (or simply worst group performance). In individual fairness, a common metric is prediction consistency, i.e., a fraction of test samples where predictions remain unchanged under certain modifications to the inputs, which maintain a similarity from the fairness standpoint.

In Toxicity, the group memberships can be defined either with respect to human annotations provided with the dataset, or with respect to the presence of certain identity tokens. Both groupings aim at highlighting comments that refer to identities that are subject to online harassment. To quantify

domain generalization, we evaluate average per group true negative (non-toxic) rate, where each group is weighted equally. We choose true negative rate (TNR) because underrepresented groups tend to have a larger fraction of toxic comments in the train data, thus being spuriously associated with toxicity by the model yielding poor TNR. This is similar to how the background is spurious in the popular domain generalization Waterbirds benchmark [15]. We weigh each group equally to ensure that performance on underrepresented groups is factored in (a more robust alternative to worst group performance). We consider both groupings, i.e., TNR (Annotations) and TNR (Identity tokens).

In Bios, the task is to predict the occupation of a person from their biography. This dataset has been mostly studied in the fairness literature [3], [7], [30], [31], but it can also be considered from the domain generalization perspective. Many of the occupations in the dataset exhibit large gender imbalance associated with historical biases, e.g., most nurses are female and most attorneys are male. Thus, gender pronouns and names can introduce spurious relations with the occupation prediction. To quantify this effect from the domain generalization perspective, we report the average of the worst accuracies with respect to the gender for each occupation (Worst per gender). Since both datasets are class-imbalanced, we also report balanced (by class) test accuracy (BA) on source to ensure that in-distribution performance remains reasonable.

**Results**   In Table 1, we compare methods for enforcing individual fairness with an ERM baseline. IF methods require a fair metric that encodes that changes in identity tokens result in similar comments in Toxicity, and changes in gender pronouns and names result in similar biographies in Bios (except for CLP which instead uses this intuition for data augmentation). We obtained the fair metric as in the original studies of the corresponding methods. We can observe that IF methods consistently improve domain generalization metrics supporting our theoretical findings. They also tend to maintain reasonable in-distribution performance, supporting their overall applicability in practical use-cases where both in- and out-of-distribution performance is important. Among the IF methods, SenSeI performs slightly better overall. We refer to Appendix F for additional results verifying that the considered methods also achieve IF.

Table 1: Enforcing domain generalization using individual fairness methods. Means and stds over 10 runs.

| | Bios | | Toxicity | | |
|---|---|---|---|---|---|
| | BA | Worst p. gender | BA | TNR (Annot.) | TNR (Id. tokens) |
| Baseline | $84.2\% \pm 0.2\%$ | $77.9\% \pm 0.4\%$ | $\mathbf{80.7}\% \pm 0.2\%$ | $79.4\% \pm 2.2\%$ | $75.0\% \pm 2.3\%$ |
| GLIF | $\mathbf{84.6}\% \pm 0.3\%$ | $77.6\% \pm 1.0\%$ | $70.5\% \pm 7.1\%$ | $\mathbf{87.0}\% \pm 9.8\%$ | $\mathbf{84.5}\% \pm 9.8\%$ |
| SenSeI | $84.3\% \pm 0.3\%$ | $\mathbf{80.2}\% \pm 0.4\%$ | $79.1\% \pm 0.5\%$ | $83.5\% \pm 1.7\%$ | $79.4\% \pm 1.5\%$ |
| SenSR | $84.2\% \pm 0.3\%$ | $\mathbf{80.2}\% \pm 0.4\%$ | $79.4\% \pm 0.3\%$ | $81.5\% \pm 1.1\%$ | $77.2\% \pm 0.9\%$ |
| CLP | $84.1\% \pm 0.3\%$ | $79.9\% \pm 0.3\%$ | $79.5\% \pm 0.6\%$ | $81.6\% \pm 1.7\%$ | $78.0\% \pm 1.8\%$ |

## 3   Individual Fairness via Domain Adaptation

In the previous section, we established that it is possible to use IF regularizers for domain adaptation problems provided that the true underlying signal satisfies some smoothness conditions. In this section, we investigate the opposite direction, i.e., whether the techniques employed for DA can be leveraged to enforce IF. Many DA methods aim at finding a representation $\Phi(X)$ of the input sample $X$, such that the source and the target distributions of $\Phi(X)$ are aligned. In other words, the goal is to make it hard to distinguish $\Phi(X_{S_i})'s$ from $\Phi(X_{T_i})'s$. For example, Ganin *et al.* [10] proposed the Domain Adversarial Neural Network (DANN) for learning $\Phi(X)$, such that the discriminator fails to discriminate between $\Phi(X_S)$ and $\Phi(X_T)$. Shu *et al.* [11] assume that the target distribution is clustered with respect to the classes and consequently the optimal classifier should pass through the low density region. To promote this condition, they modify the previous objective [10] with additional regularizers to ensure that the final classifier (which is built on top of $\Phi(X)$) has low entropy on the target and is also locally Lipschitz. Sun *et al.* [32] learn a linear transformation of the source distribution (which was later extended to learn non-linear transformations [33]), such that the first two moments of the transformed representations are the same in source and target distributions. Shen *et al.* [12] learn domain invariant representations by minimizing the Wasserstein distance between the distributions of source and target representations induced by $\Phi(X)$.

A common underlying theme of all of the above methods is to find $\Phi(X)$ which has a similar distribution on both the source and the target. In this section, we show that learning this *domain invariant* map indeed enforces individual fairness under suitable choice of domains. We demonstrate this by the following factor model: suppose we want to achieve individual fairness against a binary protected attribute $Z$ (say sex). We define two domains as two groups corresponding the protected attribute, e.g., the source domain may consist of all the observations corresponding to the males and the target domain may consist of all the observations corresponding to the females. We assume that the covariates follow a factor model structure $X = AU + bZ + \epsilon$ for three independent random variables $(U, Z, \epsilon)$ where $U$ denotes the relevant attribute, $Z$ denotes the protected attributes and $\epsilon$ is the noise. Therefore, according to our design:

$$X_S \overset{\mathscr{L}}{=} AU + b + \epsilon, \qquad (3.1) \qquad\qquad X_T \overset{\mathscr{L}}{=} AU + \epsilon. \qquad (3.2)$$

In the following theorem, we establish that if we estimate some linear transformation $\Phi \in \mathbf{R}^{q \times p}$ (with $q < p$, $p$ being the ambient dimension of $X$) of $X$ such that $\Phi X_S$ and $\Phi X_T$ has same distribution, then $\Phi b = 0$. Therefore, $\Phi X$ ignores the direction corresponding to the protected attribute and consequently is an individually fair representation.

**Theorem 3.1.** *Suppose the source and target distributions satisfy* (3.1) *and* (3.2). *If some linear transformation $\Phi X$ satisfies $\Phi X_S \overset{\mathscr{L}}{=} \Phi X_T$, then $\Phi b = 0$.*

This theorem implies any classifier built on top of the linear representation $\Phi x$ will be individually fair because $\Phi x = \Phi x'$ for any $x, x'$ that share relevant attributes $U$. The proof of the theorem can be found in the appendix. The above theorem constitutes an example of how domain adaptation methods can be adapted to enforce individual fairness when the covariates follow a factor structure.

### 3.1 Empirical Results

In this section, our goal is to train individually fair models using methods popularized in the domain adaptation (DA) literature. We experiment with DANN [10], VADA [11], and a variation of the Wasserstein-based DA (WDA) [12] discussed in Section 3. We present experimental details in Apx. F.

**Datasets and Metrics** We consider the same two datasets as in our domain generalization experiments in Section 2.4. We use prediction consistency (PC) to quantify individual fairness following prior works studying these datasets [3], [9]. For the Toxicity dataset, we modify identity tokens in the test comments and compute prediction consistency with respect to all 50 identity tokens [8]. A pair of comments that only differ in an identity token, e.g., "gay" vs "straight", are intuitively similar and should be assigned the same prediction to satisfy individual fairness. For the Bios dataset, we consider prediction consistency with respect to changes in gender pronouns and names. Such changes result in biographies that should be treated similarly.

In these experiments, we have one labeled training dataset, rather than labeled source and unlabeled target datasets typical for DA setting. As shown in Section 3, the key idea behind achieving individual fairness using DA techniques is to split the available train data into source and target domains such that aligning their representations pertains to the fairness goals. To this end, in the Bios dataset we split the train data into all-male and all-female biographies, and the Toxicity dataset we split into a domain with comments containing any of the aforementioned 50 identity tokens and a domain with comments without any identity tokens. The ERM baseline is trained on the complete training dataset.

**Results** We summarize the results in Table 2. Among the considered DA methods, WDA achieves best individual fairness improvements in terms of prediction consistency, while maintaining good balanced accuracy (BA). Comparing to a method de-

Table 2: Enforcing individual fairness using domain adaptation methods. Means and standard deviations over 10 runs.

|  | Bios | | Toxicity | |
|---|---|---|---|---|
|  | BA | PC | BA | PC |
| Baseline | **84.2**% $\pm$ 0.2% | 94.2% $\pm$ 0.1% | 80.7% $\pm$ 0.2% | 62.1% $\pm$ 1.4% |
| DANN | 84.0% $\pm$ 0.3% | 94.8% $\pm$ 0.3% | **80.8**% $\pm$ 0.2% | 62.8% $\pm$ 1.1% |
| VADA | 84.0% $\pm$ 0.3% | 94.8% $\pm$ 0.3% | **80.8**% $\pm$ 0.2% | 62.0% $\pm$ 1.4% |
| WDA | 83.3% $\pm$ 0.3% | **95.5**% $\pm$ 0.3% | 80.5% $\pm$ 0.3% | **65.4**% $\pm$ 1.3% |
| SenSeI | 84.3% $\pm$ 0.3% | 97.7% $\pm$ 0.1% | 79.1% $\pm$ 0.5% | 77.3% $\pm$ 4.3% |

signed for training individually fair models, SenSeI, prediction consistency of DA methods is worse; however, the subject understanding required to apply them is milder. Individual fairness methods require a problem-specific fair metric, which can be learned from the data, but even then requires user to define, e.g., groups of comparable samples [18]. The domain adaptation approach requires a fairness-related splitting of the train data. In our experiments, we adopted straightforward data splitting strategies and demonstrated improvements over the baseline. More sophisticated data splitting approaches can help to achieve further individual fairness improvements. We present additional experimental details in Appendix F.

## 4 Conclusion

We showed that algorithms for enforcing individual fairness (IF) can help ML models generalize to new domains and vice versa. From the lens of algorithmic fairness, the results in Section 2 show that enforcing IF can mitigate algorithmic biases caused by covariate shift *as long as the regression function satisfies IF*. This complements the recent results on mitigating algorithmic biases caused by subpopulation shift with group fairness [34]. On the other hand, compared to existing results on out-of-distribution accuracy of ML models, the results in Section 2 demonstrate the importance of inductive biases in helping models adapt to new domains. One limitation of our analysis is the assumption of covariate shift. We have relaxed this assumption in Appendix E (see Theorem E.1), where we establish results for more general distribution shifts (e.g. label shift, posterior drift etc.).

In Section 3, we showed a probabilistic connection between domain adaptation (DA) and IF. As we saw, it is possible to enforce IF by aligning the distributions of the features under a factor model. This factor model is implicit in some prior works on algorithmic fairness [18], [35], but we are not aware of any results that show it is possible to enforce IF using DA techniques.

Recent DA methods typically leverage many inductive biases through data augmentations and regularizers, and our results suggest that IF can also be leveraged. For example, utilizing annotations to identify similar images [36] can be used to learn a "fair" metric for an IF-based regularizer. We also note that our approach is similar to that of consistency regularization for DA (e.g. see [37], [38], [39], [40]) where the key idea is to ensure that *similar samples should yield similar labels*. We show that regularizer for enforcing IF can also be used as a consistency regularizer for extrapolation on the test domain. Finally, from the perspective of achieving IF, a study of different strategies for data partitioning in combination with modern DA best practices is an interesting direction for future work.

## Acknowledgments and Disclosure of Funding

This paper is based upon work supported by the National Science Foundation (NSF) under grants no. 1916271, 2027737, and 2113373 as well as the DFG in the Cluster of Excellence EXC 2117 "Centre for the Advanced Study of Collective Behaviour" (Project-ID 390829875).

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
