# A Proof of Theorems

## A.1 Proof of Theorem 2.4

For the proof of this theorem, we need few auxiliary lemmas, which we state below:

**Lemma A.1.** *Define the extrapolation map $y_t^* : \mathbf{R}^{n_s} \mapsto \mathbf{R}^{n_t}$ as:*

$$y_t^*(v) = \arg\min_{t \in \mathbf{R}^{n_t}} \mathcal{R}_n(v, t).$$

*Then under our assumptions on $\mathcal{R}_n$:*

- $y_t^*$ *is Lipschitz with Lipschitz constant $\frac{L_{\mathcal{R}}}{\mu_{\mathcal{R}}}$.*

- *For any vector $(v_s, v_t)$ we have:* $\|v_t - y_t^*(v_s)\|^2 \leq \frac{2n}{\mu_{\mathcal{R}}} \mathcal{R}(v_s, v_t)$.

**Lemma A.2.** *Under Assumption 2.3 we have:*

$$\|f(X_s) - f_0(X_s)\|_2^2 \leq \frac{2}{\mu_{\mathcal{L}}} \mathcal{L}\left(f(X_s), f_0(X_s)\right)$$

*for any function $f$. Furthermore, if $\partial_1 \mathcal{L}$ and $\partial_2 \mathcal{L}$ denotes the first and second partial derivative of $\mathcal{L}$ respectively, then we have:*

$$|\partial_1 \mathcal{L}(a, b)| \leq L_{\mathcal{L}} |a - b|,$$
$$|\partial_2 \mathcal{L}(a, b)| \leq L_{\mathcal{L}} |a - b|.$$

The proof of Lemma A.1 can be found in Section B.1 and the proof of Lemma A.2 can be found in Section B.2. For the rest of the proof, we introduce some notations for the ease of presentation: for any two vector $v_1, v_2$ of the same dimension we use $\mathcal{L}(v_1, v_2)$ or its partial derivatives to denote the coordinate wise sum, i.e., $\sum_j \mathcal{L}(v_{1,j}, v_{2,j})$. From the strong smoothness condition on $\mathcal{L}$ we have:

$$
\begin{aligned}
\frac{1}{n_t} \mathcal{L}\left(\hat{f}(X_t), f_0(X_t)\right) \leq \ & \frac{1}{n_t} \mathcal{L}\left(y_t^*(\hat{f}(X_s)), f_0(X_t)\right) \\
& + \frac{1}{n_t} \left\langle \hat{f}(X_t) - y_t^*(\hat{f}(X_s)), \partial_1 \mathcal{L}\left(y_t^*(\hat{f}(X_s)), f_0(X_t)\right) \right\rangle \\
& + \frac{L_{\mathcal{L}}}{2n_t} \left\| \hat{f}(X_t) - y_t^*(\hat{f}(X_s)) \right\|_2^2
\end{aligned}
\tag{A.1}
$$

We can further bound the first term on the RHS of the above equation as follows:

$$
\begin{aligned}
\frac{1}{n_t} \mathcal{L}\left(y_t^*(\hat{f}(X_s)), f_0(X_t)\right) \leq \ & \frac{1}{n_t} \mathcal{L}\left(y_t^*(\hat{f}(X_s)), y_t^*(f_0(X_s))\right) \\
& + \frac{1}{n_t} \left\langle f_0(X_t) - y_t^*(f_0(X_s)), \partial_2 \mathcal{L}\left(y_t^*(\hat{f}(X_s)), y_t^*(f_0(X_s))\right) \right\rangle \\
& + \frac{L_{\mathcal{L}}}{2n_t} \| f_0(X_t) - y_t^*(f_0(X_s)) \|^2
\end{aligned}
\tag{A.2}
$$

Combining the bounds on Equations (A.1) and (A.2) we obtain:

$$\frac{1}{n_t}\mathcal{L}\left(\hat{f}(X_t), f_0(X_t)\right) \le \underbrace{\frac{1}{n_t}\mathcal{L}\left(y_t^*(\hat{f}(X_s)), y_t^*(f_0(X_s))\right)}_{T_1}$$

$$+ \underbrace{\frac{1}{n_t}\left\langle f_0(X_t) - y_t^*(f_0(X_s)), \partial_2\mathcal{L}\left(y_t^*(\hat{f}(X_s)), y_t^*(f_0(X_s))\right)\right\rangle}_{T_2}$$

$$+ \underbrace{\frac{1}{n_t}\left\langle \hat{f}(X_t) - y_t^*(\hat{f}(X_s)), \partial_1\mathcal{L}\left(y_t^*(\hat{f}(X_s)), f_0(X_t)\right)\right\rangle}_{T_3}$$

$$+ \underbrace{\frac{L_{\mathcal{L}}}{2n_t}\left\| f_0(X_t) - y_t^*(f_0(X_s))\right\|^2}_{T_4}$$

$$+ \underbrace{\frac{L_{\mathcal{L}}}{2n_t}\left\| \hat{f}(X_t) - y_t^*(\hat{f}(X_s))\right\|^2}_{T_5} \tag{A.3}$$

The term $T_4, T_5$ can be bounded directly by Lemma A.1 as:

$$T_4 \le \frac{L_{\mathcal{L}}\, n}{\mu_{\mathcal{R}}\, n_t}\mathcal{R}\left(f_0(X_s), f_0(X_t)\right) \tag{A.4}$$

$$T_5 \le \frac{L_{\mathcal{L}}\, n}{\mu_{\mathcal{R}}\, n_t}\mathcal{R}\left(\hat{f}(X_s), \hat{f}(X_t)\right) \tag{A.5}$$

To bound $T_2$, using $ab \le (a^2 + b^2)/2$ we have:

$$T_2 = \frac{1}{n_t}\left\langle f_0(X_t) - y_t^*(f_0(X_s)), \partial_2\mathcal{L}\left(y_t^*(\hat{f}(X_s)), y_t^*(f_0(X_s))\right)\right\rangle$$

$$\le \frac{1}{n_t}\left\| f_0(X_t) - y_t^*(f_0(X_s))\right\|^2 + \frac{1}{n_t}\left\|\partial_2\mathcal{L}\left(y_t^*(\hat{f}(X_s)), y_t^*(f_0(X_s))\right)\right\|^2$$

$$\le \frac{2n}{\mu_{\mathcal{R}}\, n_t}\mathcal{R}\left(f_0(X_s), f_0(X_t)\right) + \frac{L_{\mathcal{L}}^2}{4n_t}\left\| y_t^*(\hat{f}(X_s)) - y_t^*(f_0(X_s))\right\|^2 \quad [\text{Lemma } A.2]$$

$$\le \frac{2n}{\mu_{\mathcal{R}}\, n_t}\mathcal{R}\left(f_0(X_s), f_0(X_t)\right) + \frac{L_{\mathcal{L}}^2 L_{\mathcal{R}}^2}{4\mu_{\mathcal{R}}^2\, n_t}\left\| \hat{f}(X_s) - f_0(X_s)\right\|^2 \quad [\text{Lemma } A.1]$$

$$\le \frac{2n}{\mu_{\mathcal{R}}\, n_t}\mathcal{R}\left(f_0(X_s), f_0(X_t)\right) + \frac{L_{\mathcal{L}}^2 L_{\mathcal{R}}^2}{2\mu_{\mathcal{R}}^2\mu_{\mathcal{L}}} \times \frac{n_s}{n_t} \times \frac{1}{n_s}\mathcal{L}\left(\hat{f}(X_s), f_0(X_s)\right) \quad [\text{Lemma } A.2]$$

The bound on $T_3$ follows from a similar line of argument:

$$T_3 = \frac{1}{n_t}\left\langle \hat{f}(X_t) - y_t^*(\hat{f}(X_s)), \partial_1\mathcal{L}\left(y_t^*(\hat{f}(X_s)), f_0(X_t)\right)\right\rangle$$

$$\le \frac{1}{n_t}\left\| \hat{f}(X_t) - y_t^*(\hat{f}(X_s))\right\|^2 + \frac{1}{n_t}\left\|\partial_1\mathcal{L}\left(y_t^*(\hat{f}(X_s)), f_0(X_t)\right)\right\|^2$$

$$\le \frac{2n}{\mu_{\mathcal{R}}\, n_t}\mathcal{R}\left(\hat{f}(X_s), \hat{f}(X_t)\right) + \frac{L_{\mathcal{L}}^2}{4n_t}\left\| y_t^*(\hat{f}(X_s)) - f_0(X_t)\right\|^2 \quad [\text{Lemma } A.2]$$

$$\le \frac{2n}{\mu_{\mathcal{R}}\, n_t}\mathcal{R}\left(\hat{f}(X_s), \hat{f}(X_t)\right) + \frac{L_{\mathcal{L}}^2}{2n_t}\left\| y_t^*(\hat{f}(X_s)) - y_t^*(f_0(X_s))\right\|^2 + \frac{L_{\mathcal{L}}^2}{2n_t}\left\| y_t^*(f_0(X_s)) - f_0(X_t)\right\|^2$$

$$\le \frac{2n}{\mu_{\mathcal{R}}\, n_t}\mathcal{R}\left(\hat{f}(X_s), \hat{f}(X_t)\right) + \frac{L_{\mathcal{L}}^2\, n}{\mu_{\mathcal{R}}\, n_t}\mathcal{R}\left(f_0(X_s), f_0(X_t)\right) +$$

$$+ \frac{L_{\mathcal{L}}^2 L_{\mathcal{R}}^2}{\mu_{\mathcal{R}}^2\mu_{\mathcal{L}}} \times \frac{n_s}{n_t} \times \frac{1}{n_s}\mathcal{L}\left(\hat{f}(X_s), f_0(X_s)\right) \quad [\text{Lemma } A.2]$$

Finally to bound $T_1$ we use again Assumption 2.3, i.e., strong convexity and strong smoothness of $\mathcal{L}$ as follows:

$$
\begin{aligned}
T_1 &= \frac{1}{n_t} \mathcal{L}\left(y_t^*(\hat{f}(X_s)), y_t^*(f_0(X_s))\right) \\
&\leq \frac{L_{\mathcal{L}}}{2n_t} \left\| y_t^*(\hat{f}(X_s)) - y_t^*(f_0(X_s)) \right\|_2^2 \\
&\leq \frac{L_{\mathcal{L}} L_{\mathcal{R}}^2}{2\mu_{\mathcal{R}}^2 \, n_t} \left\| \hat{f}(X_s) - f_0(X_s) \right\|_2^2 \\
&\leq \frac{L_{\mathcal{L}} L_{\mathcal{R}}^2}{2\mu_{\mathcal{R}}^2} \times \frac{n_s}{n_t} \times \frac{1}{n_s} \mathcal{L}\left(\hat{f}(X_s), f_0(X_s)\right)
\end{aligned}
$$

Suppose $\rho_n = n_s/n_t$. Then we have $n/n_t = 1 + \rho_n$. Using this notation and combining the bound on all $\{T_i\}_{i=1}^5$, we obtain:

$$
\begin{aligned}
\frac{1}{n_t} \mathcal{L}\left(\hat{f}(X_t), f_0(X_t)\right) &\leq \frac{L_{\mathcal{L}} L_{\mathcal{R}}^2 (\mu_{\mathcal{L}} + 3L_{\mathcal{L}})}{2\mu_{\mathcal{R}}^2 \mu_{\mathcal{L}}} \rho_n \frac{1}{n_s} \mathcal{L}\left(\hat{f}(\mathbf{X}_s), f_0(\mathbf{X}_s)\right) \\
&\quad + \frac{2 + L_{\mathcal{L}}}{\mu_{\mathcal{R}}} (1 + \rho_n) \mathcal{R}(\hat{f}(\mathbf{X})) + \frac{2 + L_{\mathcal{L}} + L_{\mathcal{L}}^2}{\mu_{\mathcal{R}}} (1 + \rho_n) \mathcal{R}(f_0(\mathbf{X})) \\
&\leq \alpha_n \left[ \frac{1}{n_s} \mathcal{L}\left(\hat{f}(\mathbf{X}_s), f_0(\mathbf{X}_s)\right) + \lambda \mathcal{R}(\hat{f}(\mathbf{X})) \right] + \beta_n \mathcal{R}(f_0(\mathbf{X})),
\end{aligned} \tag{A.6}
$$

with the values of $\alpha_n$ and $\beta_n$ being:

$$
\alpha_n = \max\left\{ \frac{L_{\mathcal{L}} L_{\mathcal{R}}^2 (\mu_{\mathcal{L}} + 3L_{\mathcal{L}})}{2\mu_{\mathcal{R}}^2 \mu_{\mathcal{L}}} \rho_n, \frac{2 + L_{\mathcal{L}}}{\lambda \mu_{\mathcal{R}}} (1 + \rho_n) \right\}, \tag{A.7}
$$

$$
\beta_n = \frac{2 + L_{\mathcal{L}} + L_{\mathcal{L}}^2}{\mu_{\mathcal{R}}} (1 + \rho_n). \tag{A.8}
$$

This completes the proof.

## A.2 Proof of Theorem 2.7

First, note that, from Assumption 2.3 we have:

$$
\begin{aligned}
&\mathbb{E}_Q\left[\mathcal{L}(\tilde{f}(x), f_0(x))\right] \\
&\leq \mathbb{E}_Q[\mathcal{L}(f_0(x), f_0(x))]^{\nearrow 0} + \mathbb{E}\left[(\tilde{f}(x) - f_0(x))\partial_1 \mathcal{L}(f_0(x), f_0(x))\right]^{\nearrow 0} + \frac{L_{\mathcal{L}}}{2} \left\| \tilde{f}(x) - f_0(x) \right\|_Q^2 \\
&= \frac{L_{\mathcal{L}}}{2} \left\| \hat{f}(x) - f_0(x) \right\|_Q^2 .
\end{aligned} \tag{A.9}
$$

and

$$
\begin{aligned}
&\mathbb{E}_P\left[\mathcal{L}(\tilde{f}(x), f_0(x))\right] \\
&\geq \mathbb{E}_P[\mathcal{L}(f_0(x), f_0(x))]^{\nearrow 0} + \mathbb{E}_P\left[(\tilde{f}(x) - f_0(x))\partial_1 \mathcal{L}(f_0(x), f_0(x))\right]^{\nearrow 0} + \frac{\mu_{\mathcal{L}}}{2} \left\| \tilde{f}(x) - f_0(x) \right\|_P^2 \\
&= \frac{\mu_{\mathcal{L}}}{2} \left\| \hat{f}(x) - f_0(x) \right\|_P^2 .
\end{aligned} \tag{A.10}
$$

Therefore, it is enough to bound $\|\hat{f}(x) - f_0(x)\|_Q^2$. As per Assumption 2.6, $\mathcal{R}$ is strongly convex with respect to its second coordinate, i.e.,

$$
\mathcal{R}(f, g) \geq \mathcal{R}(f, \tilde{g}) + \partial_2 \mathcal{R}((f, g); g - \tilde{g}) + \frac{\mu_{\mathcal{R}}}{2} \|g - \tilde{g}\|_Q^2 .
$$

We now define an operator $M$ along the line of $y_t^*$ as $M(f) = \arg\min_g \mathcal{R}(f, g)$. As $M(f)$ is the minimizer over of the second coordinate, we have $\partial_2 R(f, M(f)) = 0$ and consequently from the strong convexity of $R$ we have:

$$\mathcal{R}(f, f) \geq \mathcal{R}(f, M(f)) + \frac{\mu_{\mathcal{R}}}{2} \|f - M(f)\|_Q^2 \ .$$

The above inequality implies:

$$\|f - M(f)\|_Q^2 \leq \frac{2}{\mu_L} \left[\mathcal{R}(f, f) - \mathcal{R}(f, M(f))\right] \leq \frac{2}{\mu_L} \mathcal{R}(f, f) \ .$$

which will be used later in our proof.

$M$ **is Lipschitz:** By definition of $M$ we have $\partial_2 \mathcal{R}(f, M(f)) = 0$, which further implies for any two functions $f_1, f_2$:

$$0 = \partial_2 \mathcal{R}((f_1, M(f_1)); (M(f_1) - M(f_2))) - \partial_2 \mathcal{R}((f_2, M(f_2)); (M(f_1) - M(f_2)))$$
$$= \partial_2 \mathcal{R}((f_1, M(f_1)); (M(f_1) - M(f_2))) - \partial_2 \mathcal{R}((f_1, M(f_2)); (M(f_1) - M(f_2)))$$
$$+ \partial_2 \mathcal{R}((f_1, M(f_2)); (M(f_1) - M(f_2))) - \partial_2 \mathcal{R}((f_2, M(f_2)); (M(f_1) - M(f_2)))$$

Changing side we obtain:

$$\partial_2 \mathcal{R}((f_2, M(f_2)); (M(f_1) - M(f_2))) - \partial_2 \mathcal{R}((f_1, M(f_2)); (M(f_1) - M(f_2)))$$
$$= \partial_2 \mathcal{R}((f_1, M(f_1)); (M(f_1) - M(f_2))) - \partial_2 \mathcal{R}((f_1, M(f_2)); (M(f_1) - M(f_2)))$$
$$\geq \mu_{\mathcal{R}} \|M(f_1) - M(f_2)\|_Q^2 \tag{A.11}$$

where the last inequality follows from the strong convexity of $\mathcal{R}$ (Assumption 2.6). Furthermore, we have:

$$\partial_2 \mathcal{R}((f_2, M(f_2)); (M(f_1) - M(f_2))) - \partial_2 \mathcal{R}((f_1, M(f_2)); (M(f_1) - M(f_2)))$$
$$\leq L_{\mathcal{R}} \|f_1 - f_2\|_P \|M(f_1) - M(f_2)\|_Q \ . \tag{A.12}$$

This follows from the second part of Assumption 2.6. Combining Equation (A.11) and (A.12), we conclude:

$$\|M(f_1) - M(f_2)\|_Q \leq \frac{L_{\mathcal{R}}}{\mu_{\mathcal{R}}} \|f_1 - f_2\|_P \ .$$

We now return to the main proof:

$$\left\|\tilde{f}_Q - f_0\right\|_Q^2 \leq \left\|\tilde{f}_Q - M(\tilde{f}_Q)\right\|_Q^2 + \left\|M(\tilde{f}_Q) - M(f_0)\right\|_Q^2 + \|f_0 - M(f_0)\|_Q^2$$
$$\leq \frac{2}{\mu_{\mathcal{R}}} \left(\mathcal{R}(f_0) + \mathcal{R}(\tilde{f}_Q)\right) + \frac{L_{\mathcal{R}}}{\mu_{\mathcal{R}}} \left\|\tilde{f}_Q - f_0\right\|_P^2$$
$$:= C_0 \left[\left\|\tilde{f}_Q - f_0\right\|_P^2 + \lambda \mathcal{R}(\tilde{f}_Q)\right] + C_2 \mathcal{R}(f_0)$$
$$\leq C_1 \left[\mathbb{E}_P \left[\mathcal{L}(\tilde{f}(x), f_0(x))\right] + \lambda \mathcal{R}(\tilde{f}_Q)\right] + C_2 \mathcal{R}(f_0)$$

where the first term on the right hand side is the minimum training error (population version, i.e., in presence of infinite sample) and the second term quantifies the smoothness of $f_0$ in terms of the regularizer $R$. The last inequality follows from the strong convexity of the loss function ((A.10)).

## A.3  Proof of Theorem 2.9

In this section, we prove Theorem 2.9. Fix $Q \in \mathcal{Q}_\epsilon$. Then there exists some $T \equiv T(Q) \in \mathcal{T}_\epsilon$ such that $T \# P = Q$. Define an operator $M_T$ as:

$$M_T(f) = \arg\min_g \mathcal{R}_T(f, g)$$

where $\mathcal{R}_T(f, g) = \mathbb{E}_{x \sim P} \left[(f(x) - g(T(x)))^2\right]$. The proof of the strong convexity of $R_T$ with respect to its second coordinate is straightforward as we have the following double Gateaux derivative:

$$\partial_2^2 \mathcal{R}((f, g) : h_1, h_2) = 2\mathbb{E}_{x \sim P} \left[h_1(T(x))h_2(T(x))\right] \ .$$

Fix $f \in \mathcal{F}$ and define $\Delta = f \circ T - M_T(f) \circ T$. A two step Taylor expansion yields:

$$
\begin{aligned}
\mathcal{R}_T(f, f) &= \mathcal{R}_T(f, M_T(f)) + \partial_2 \mathcal{R}_T((f, M_T(f)); \Delta) + \frac{1}{2} \partial_2 \mathcal{R}_T((f, f^*); \Delta, \Delta) \\
&= \mathcal{R}_T(f, M_T(f)) + \mathbb{E}[\Delta^2] \\
&= \mathcal{R}_T(f, M_T(f)) + \|f - M_T(f)\|_Q^2 .
\end{aligned}
$$

where the derivative is canceled because $M_T(f)$ is the minimizer. Therefore, we have:

$$
\|f - M_T(f)\|_Q^2 = \mathcal{R}_T(f, f) - \mathcal{R}_T(f, M_T(f)) \leq \mathcal{R}_T(f, f) . \tag{A.13}
$$

We use the above bound in our subsequent calculation:

$$
\begin{aligned}
\left\| \tilde{f} - f_0 \right\|_Q^2 &\leq 4 \left[ \left\| \tilde{f} - M_T(\tilde{f}) \right\|_Q^2 + \left\| M_T(\tilde{f}) - M_T(f_0) \right\|_Q^2 + \|f_0 - M_T(f_0)\|_Q^2 \right] \\
&\leq 4 \left[ \mathcal{R}_T(\tilde{f}, \tilde{f}) + \left\| M_T(\tilde{f}) - M_T(f_0) \right\|_Q^2 + \mathcal{R}_T(f_0, f_0) \right] \quad \text{[From (A.13)]} \tag{A.14}
\end{aligned}
$$

We now bound the second term of the RHS of the above equation. Following the similar calculation as in (A.11) and (A.12) we have for any function $f_1, f_2$:

$$
\|M(f_1) - M(f_2)\|_Q \leq \|f_1 - f_2\|_P .
$$

In particular for $f_1 = \tilde{f}$ and $f_2 = f_0$ we have:

$$
\left\| M(\tilde{f}) - M(f_0) \right\|_Q \leq \left\| \tilde{f} - f_0 \right\|_P . \tag{A.15}
$$

Combining the bound in (A.14) and (A.15) we conclude that for any $Q \in \mathbb{Q}_\epsilon$:

$$
\left\| \tilde{f} - f_0 \right\|_Q^2 \leq 4 \left[ R_T(\tilde{f}, \tilde{f}) + R_T(f_0, f_0) + \left\| \tilde{f} - f_0 \right\|_P^2 \right]
$$

Taking the supremum with respect to $Q$ on both sides, we conclude the proof of the theorem.

### A.4 Proof of Theorem 3.1

The proof follows from analyzing the characteristic function of $X_s$ and $X_t$. Note that by definition:

$$
\begin{aligned}
\phi_{\Phi X_s}(t) &= \mathbb{E} \left[ e^{it^\top \Phi X_s} \right] \\
&= \mathbb{E} \left[ e^{it^\top (\Phi A U + \Phi b + \Phi \epsilon)} \right] \\
&= \phi_U(A^\top \Phi^\top t) \, \phi_\epsilon(\Phi^\top t) \, e^{it^\top \Phi b}
\end{aligned}
$$

Similarly, for $X_t$ we have:

$$
\phi_{\Psi X_t}(t) = \mathbb{E} \left[ e^{it^\top (\Phi A U + \Phi \epsilon)} \right] = \phi_U(A^\top \Phi^\top t) \, \phi_\epsilon(\Phi^\top t) = \phi_{\Phi X_s}(t) \, e^{it^\top \Phi b} .
$$

Therefore, if $\Phi X_s \overset{\mathscr{L}}{=} \Phi X_t$, $\phi_{\Psi X_t}(t) = \phi_{\Phi X_s}(t)$ for all $t$, which further implies $e^{it^\top \Phi b} = 1$ for all $t$, which implies $\Phi b = 0$. This completes the proof.

## B  Proof of Auxiliary Lemmas

### B.1  Proof of Lemma A.1

The proof of the second part of the above lemma follows directly from the strong convexity of $\mathcal{R}_n$ with respect to the second coordinate, as the strong convexity assumption yields:

$$
\mathcal{R}(v_s, v_t) \geq \mathcal{R}(v_s, y_t^*(v_s)) + \langle v_t - y_t^*(v_s), \partial_t \mathcal{R}(v_s, y_t^*(v_s)) \rangle + \frac{\mu_{\mathcal{R}}}{2n} \|v_t - y_t^*(v_s)\|^2 .
$$

The second term of the RHS of the above equation is 0 as $\partial_t \mathcal{R}(v_s, y_t^*(v_s)) = 0$ (as the derivative of a smooth function is 0 at minima). Therefore, changing sides of the terms, we conclude:

$$\|v_s - y_t^*(v_s)\|^2 \leq \frac{2n}{\mu_{\mathcal{R}}} \left( \mathcal{R}(v_s, v_t) - \mathcal{R}(v_s, y_t^*(v_s)) \right) \leq \frac{2n}{\mu_{\mathcal{R}}} \mathcal{R}(v_s, v_t)$$

where the last inequality follows from the non-negativity of $\mathcal{R}_n$. This completes the proof of the second part of the lemma.

For the first part of the lemma, first note that we have :

$$\langle y_t^*(v_2) - y_t^*(v_1), \partial_t \mathcal{R}_n(v_1, y_t^*(v_1)) - \partial_t \mathcal{R}_n(v_2, y_t^*(v_2)) \rangle = 0$$

as $\partial_t \mathcal{R}_n(v_1, y_t^*(v_1)) = \partial_t \mathcal{R}_n(v_2, y_t^*(v_2)) = 0$ (derivative is 0 at minima). Adding and subtracting $\partial_t \mathcal{R}_n(v_1, y_t^*(v_2))$ from the above equation yields:

$$\langle y_t^*(v_2) - y_t^*(v_1), \partial_t \mathcal{R}_n(v_1, y_t^*(v_1)) - \partial_t \mathcal{R}_n(v_1, y_t^*(v_2))$$
$$+ \partial_t \mathcal{R}_n(v_1, y_t^*(v_2)) - \partial_t \mathcal{R}_n(v_2, y_t^*(v_2)) \rangle = 0$$

Changing sides, we have:

$$\langle y_t^*(v_2) - y_t^*(v_1), \partial_t \mathcal{R}_n(v_1, y_t^*(v_2)) - \partial_t \mathcal{R}_n(v_2, y_t^*(v_2)) \rangle$$
$$\geq \langle y_t^*(v_2) - y_t^*(v_1), \partial_t \mathcal{R}_n(v_1, y_t^*(v_2)) - \partial_t \mathcal{R}_n(v_1, y_t^*(v_1)) \rangle$$
$$\geq \frac{\mu_{\mathcal{R}}}{2n} \|y_t^*(v_2) - y_t^*(v_1)\|^2 \ . \tag{B.1}$$

On the other hand, a simple application of the Cauchy-Schwarz inequality yields:

$$\langle y_t^*(v_2) - y_t^*(v_1), \partial_t \mathcal{R}_n(v_1, y_t^*(v_2)) - \partial_t \mathcal{R}_n(v_2, y_t^*(v_2)) \rangle$$
$$\leq \|y_t^*(v_2) - y_t^*(v_1)\| \, \|\partial_t \mathcal{R}_n(v_1, y_t^*(v_2)) - \partial_t \mathcal{R}_n(v_2, y_t^*(v_2))\|$$
$$\leq \frac{L_{\mathcal{R}}}{2n} \|y_t^*(v_2) - y_t^*(v_1)\| \, \|v_1 - v_2\| \ . \tag{B.2}$$

Combining the bounds of Equation (B.1) and (B.2), we have:

$$\|y_t^*(v_2) - y_t^*(v_1)\| \leq \frac{L_{\mathcal{R}}}{\mu_{\mathcal{R}}} \|v_1 - v_2\| \ ,$$

which completes the proof.

## B.2 Proof of Lemma A.2

The proof follows directly from the following properties of the $\mathcal{L}$:

1. $\mathcal{L}(f_0(X_s), f_0(X_s)) = 0$.
2. $\partial_1 \mathcal{L}(f_0(X_s), f_0(X_s)) = \partial_2 \mathcal{L}(f_0(X_s), f_0(X_s)) = 0$
3. $\mathcal{L}$ is strongly convex.

From strong convexity of $\mathcal{L}$ we have:

$$\mathcal{L}\left(\hat{f}(X_s), f_0(X_s)\right) \geq \mathcal{L}\left(f_0(X_s), f_0(X_s)\right)$$
$$+ \left\langle \hat{f}(X_s) - f_0(X_s), \partial_1 \mathcal{L}\left(f_0(X_s), f_0(X_s)\right) \right\rangle$$
$$+ \frac{\mu_{\mathcal{L}}}{2} \left\| \hat{f}(X_s) - f_0(X_s) \right\|^2$$

The first and second term on the RHS will be 0 by the first and second properties of $\mathcal{L}$ mentioned above. Therefore, we have:

$$\mathcal{L}\left(\hat{f}(X_s), f_0(X_s)\right) \geq \frac{\mu_{\mathcal{L}}}{2} \left\| \hat{f}(X_s) - f_0(X_s) \right\|^2$$

which completes the proof.

## C Similarity Kernel-based Regularizer

A similarity kernel-based regularizer $\mathcal{R}$ is defined as:

$$\mathcal{R}(f, g) = \mathbb{E}_{\substack{X \sim P \\ X' \sim Q}} \left[ (f(X) - g(X'))^2 K(X, X') \right]$$

where $K$ is the kernel of similarity. In particular, if an $x$ from the source domain is *similar* to an $x'$ in the target domain in the sense that $f_0(x) \approx f_0(x')$, then we expect the value of $K(x, x')$ to be large. In this section, we show that under some mild regularity condition on $K$, this regularizer satisfies Assumption 2.2.

**Assumption C.1** (Assumption on kernel). *Define $K_Q(x') = \mathbb{E}_{x \sim P}[K(x, x')]$ and $K_{\max} = \max_{x,x'} K(x, x')$. Assume that $K_{\max} < \infty$ and*

$$\inf_h \frac{\|h\sqrt{K_Q}\|_Q}{\|h\|_Q} \geq \phi > 0 \,.$$

**Gateaux derivatives of $\mathcal{R}(f, g)$:** The first order Gateaux derivative of $\mathcal{R}$ in the direction of a function $h$ is defined as:

$$
\begin{aligned}
\partial_2 \mathcal{R}((f, g); h) &= \lim_{t \downarrow 0} \frac{\mathcal{R}(f, g + th) - \mathcal{R}(f, g)}{t} \\
&= 2\mathbb{E}_{\substack{X \sim P \\ X' \sim Q}} \left[ (g(X') - f(X)) h(X') K(X, X') \right]
\end{aligned}
$$

Similarly, the second order Gateaux derivative at direction $(h_1, h_2)$ is defined as:

$$
\begin{aligned}
\partial_2^2 \mathcal{R}((f, g); h_1, h_2) &= \lim_{t \downarrow 0} \frac{\partial_2 \mathcal{R}((f, g + th_2); h_1) - \partial_2 \mathcal{R}((f, g); h_1)}{t} \\
&= 2\mathbb{E}_{\substack{X \sim P \\ X' \sim Q}} \left[ h_1(X') h_2(X') K_Q(X') \right]
\end{aligned}
$$

where $K_Q(X') = \mathbb{E}_{X \sim P}[K(X, X')]$. Therefore, the strong convexity follows from Assumption C.1.

We next show that $\mathcal{R}$ also satisfies the second condition of Assumption 2.6. Towards that direction:

$$\partial_2 R((f_2, M(f_2)); (M(f_1) - M(f_2))) - \partial_2 R((f_1, M(f_2)); (M(f_1) - M(f_2)))$$

$$
\begin{aligned}
&= 2\mathbb{E}_{\substack{X \sim P \\ X' \sim Q}} \left[ (f_1(X) - f_2(X))(M(f_1)(X') - M(f_2)(X')) K(X, X') \right] \\
&\leq K_{\max} \|f_1 - f_2\|_P \|M(f_1) - M(f_2)\|_Q \,.
\end{aligned}
$$

This concludes that the similarity kernel-based population regularizer $\mathcal{R}$ satisfies Assumption 2.6 under Assumption C.1 on the kernel function.

## D Population and Sample Version of the Regularizer

In this section, we show that under a fairly general condition, if $\mathcal{R}_n$ (the sample version of the regularization) satisfies Assumptions 2.1 and 2.2 and $\mathcal{R}$ is the asymptotic limit of $\mathcal{R}_n$, i.e., $\mathcal{R}_n \overset{a.s.}{\to} \mathcal{R}$ as $n_s, n_t \to \infty$, then $\mathcal{R}$ will satisfy Assumption 2.5 and 2.6. Towards that if $\mathcal{R}_n$ satisfies Assumption 2.1 for all $n$, then taking the limit $n \to \infty$, it is immediate that $\mathcal{R}$ satisfies Assumption 2.5.

For the other assumption, suppose $\mathcal{R}_n$ satisfies the first part of Assumption 2.2, i.e., it is strongly convex with respect to its second coordinates (the coordinates corresponding to the target samples), then again, simply taking the limit $n \to \infty$, we conclude that $\mathcal{R}$ is also strongly convex with $\mu_{\mathcal{R}} = \liminf_{n \to \infty} \mu_{\mathcal{R}_n}$ (as long as $\mu_{\mathcal{R}} > 0$). By similar argument, the second part of Assumption 2.6 is also satisfied if $\mathcal{R}_n$ satisfies the strong smoothness assumption and $\mathcal{L}_{\mathcal{R}_n}$ does not diverge to infinity.

# E  Bound for Non-Covariate Shift

In this section, we extend the result of Theorem 2.7 to the setup when the mean function $f_0$ is different on source and target domain. More precisely, we assume the following data generative process:

$$y_s = f_s(x_s) + \epsilon_s, \;\; y_t = f_t(x_t) + \epsilon_t \,. \tag{E.1}$$

The following theorem extends the bounds obtained in Theorem 2.7 for the estimator obtained via (2.3):

**Theorem E.1.** *Suppose we observe* $(Y_1, X_1), \ldots, (X_n, Y_n)$ *from the source domain and* $\tilde{X}_1, \ldots, \tilde{X}_n$ *from the target domain. The estimator* $\tilde{f}$ *obtained via Equation* (2.3) *satisfied the following generalization error bound on the target domain:*

$$\mathbb{E}_Q\left[\mathcal{L}(\tilde{f}(x), f_t(x))\right] \leq C_1 \left[\mathbb{E}_P\left[\mathcal{L}(\tilde{f}(x), f_s(x))\right] + \lambda \mathcal{R}(\tilde{f})\right]$$
$$+ C_2 \min\left\{\mathcal{R}(f_t) + \|f_s - f_t\|_P^2, \mathcal{R}(f_s) + \|f_s - f_t\|_Q^2\right\},$$

*for some constants* $C_1, C_2$ *mentioned explicitly in the proof.*

*Proof.* The proof is quite similar to the proof of Theorem 2.7, hence we will only highlight here the key difference for the sake of brevity. From the proof of Theorem 2.7 we have:

$$\mathbb{E}_Q\left[\mathcal{L}(\tilde{f}(x), f_t(x))\right] \leq \frac{L_{\mathcal{L}}}{2}\left\|\tilde{f}(x) - f_t(x)\right\|_Q^2, \tag{E.2}$$

$$\mathbb{E}_P\left[\mathcal{L}(\tilde{f}(x), f_s(x))\right] \geq \frac{\mu_{\mathcal{L}}}{2}\left\|\tilde{f}(x) - f_s(x)\right\|_P^2 \tag{E.3}$$

$$\|f - M(f)\|_Q^2 \leq \frac{2}{\mu_L}\left[\mathcal{R}(f, f) - \mathcal{R}(f, M(f))\right] \leq \frac{2}{\mu_L}\mathcal{R}(f, f), \tag{E.4}$$

$$\|M(f_1) - M(f_2)\|_Q \leq \frac{L_{\mathcal{R}}}{\mu_{\mathcal{R}}}\|f_1 - f_2\|_P. \tag{E.5}$$

An application of triangle inequality yields:

$$\left\|\tilde{f} - f_t\right\|_Q^2 \leq 8\left[\left\|\tilde{f} - M(\tilde{f})\right\|_Q^2 + \left\|M(\tilde{f}) - M(f_s)\right\|_Q^2 + \|M(f_s) - M(f_t)\|_Q^2 + \|M(f_t) - f_t\|_Q^2\right]$$

$$\leq \frac{16}{\mu_{\mathcal{R}}}\left(\mathcal{R}(f_t) + \mathcal{R}(\tilde{f})\right) + \frac{8L_{\mathcal{R}}}{\mu_{\mathcal{R}}}\left(\left\|\tilde{f} - f_s\right\|_P^2 + \|f_s - f_t\|_P^2\right)$$

$$:= \bar{C}_0\left[\left\|\tilde{f} - f_s\right\|_P^2 + \lambda \mathcal{R}(\tilde{f})\right] + \bar{C}_2 \mathcal{R}(f_t) + \bar{C}_3\|f_s - f_t\|_P^2$$

$$\leq \bar{C}_1\left[\mathbb{E}_P\left[\mathcal{L}(\tilde{f}(x), f_s(x))\right] + \lambda \mathcal{R}(\tilde{f})\right] + \bar{C}_2 \mathcal{R}(f_t) + \bar{C}_3\|f_s - f_t\|_P^2 \tag{E.6}$$

where the first term on the right hand side is the minimum training error (population version, i.e., in presence of infinite sample) and the second term quantifies the smoothness of $f_0$ in terms of the regularizer $R$. The last inequality follows from the strong convexity of the loss function (A.10). Another version of telescoping sum yields:

$$\left\|\tilde{f} - f_t\right\|_Q^2 \leq 8\left[\left\|\tilde{f} - M(\tilde{f})\right\|_Q^2 + \left\|M(\tilde{f}) - M(f_s)\right\|_Q^2 + \|M(f_s) - f_s\|_Q^2 + \|f_s - f_t\|_Q^2\right]$$

$$\leq \frac{16}{\mu_{\mathcal{R}}}\left(\mathcal{R}(f_s) + \mathcal{R}(\tilde{f})\right) + \frac{8L_{\mathcal{R}}}{\mu_{\mathcal{R}}}\left\|\tilde{f} - f_s\right\|_P^2 + 8\|f_s - f_t\|_Q^2$$

$$:= \tilde{C}_0\left[\left\|\tilde{f} - f_s\right\|_P^2 + \lambda \mathcal{R}(\tilde{f})\right] + \tilde{C}_2 \mathcal{R}(f_s) + \tilde{C}_3\|f_s - f_t\|_Q^2$$

$$\leq \tilde{C}_1\left[\mathbb{E}_P\left[\mathcal{L}(\tilde{f}(x), f_s(x))\right] + \lambda \mathcal{R}(\tilde{f})\right] + \tilde{C}_2 \mathcal{R}(f_s) + \tilde{C}_3\|f_s - f_t\|_Q^2 \tag{E.7}$$

Therefore, combining Equations (E.6) and (E.7) yields the result of the theorem. $\qquad \square$

## F   Experimental Details

In Table 3 we compare prediction consistency [3], [14] of the methods compared in Table 1 of the main text to verify that they also achieve individual fairness as intended.

Table 3: Comparison of prediction consistency in the experiment corresponding to Table 1.

|          | Bios          | Toxicity      |
|----------|---------------|---------------|
| Baseline | 94.2%±0.1%    | 62.1%±1.4%    |
| GLIF     | **98.8**%±0.2%| **84.4**%±1.3%|
| SenSeI   | 97.7%±0.1%    | 77.3%±4.3%    |
| SenSR    | 97.6%±0.1%    | 72.9%±4.4%    |
| CLP      | 97.4%±0.1%    | 76.3%±4.8%    |

We summarize some additional details regarding the implementation of domain adaptation methods in the experiments in Section 3.1.

- Since the target domains are labeled (they consist of labeled samples from the train data), we also add a loss term to the objective corresponding to the target domain performance when training the domain adaptation methods. Recall that the main mechanisms for achieving individual fairness are the representation alignment regularizers, thus adding loss in the target domain is simply a way to utilize the available labels to improve performance.

- For DANN, we use a ReLU-activated two-layer base model with 2000 hidden neurons and 768 output neurons. Further, we use a ReLU-activated two-layer base model with 100 hidden neurons and one logistically activated neuron as the discriminator. As the prediction head, we use a ReLU-activated two-layer model with 2000 hidden neurons.

- For VADA, we use the same models as for DANN, and the primary difference is the additional virtual adversarial training (VAT) loss.

- For WDA, we replaced the Wasserstein distance utilized by Shen *et al.* [12] with the Sinkhorn divergence [41]. The Sinkhorn divergence is a computationally more efficient analogue of the Wasserstein distance regularizer. We used the Geomloss package [42] in our code.

## G   Background on Domain Adaptation and Algorithmic Fairness

Domain adaptation generally refers to the problem of semi-supervised learning under distribution shift. More precisely, in the semi-supervised setting the learner is given a labeled dataset $\{(X_i, Y_i)\}_{i=1}^{n}$ and an unlabeled dataset $\{X_i\}_{i=n+1}^{m}$. In domain adaptation, we typically assume the labeled samples and unlabeled samples are drawn from a source $P$ and target distribution $Q$ that are similar but non-identical. The goal of the learner is to find a prediction rule $f : \mathcal{X} \to \mathcal{Y}$ such that $\mathbf{E}_Q\big[\ell(f(X), Y)\big]$ is small. This goal is impossible without additional assumptions restricting the differences between $P$ and $Q$. In light of the available data, a natural assumption is covariate shift: $\mathbf{E}_P\big[Y \mid X = x\big] = \mathbf{E}_Q\big[Y \mid X = x\big]$. The standard approach to this problem is importance weighing [43]. It is based on the observation that

$$\mathbf{E}_Q\big[\ell(f(X), Y))\big] = \mathbf{E}_P\big[w(X)\ell(f(X), Y)\big]], \tag{G.1}$$

where $w(x) \triangleq \frac{dQ_X}{dP_X}(x)$ is the likelihood ratio between the marginal distribution of inputs in the target and that in the source domains. It is possible to estimate $w$ from the inputs in the labeled and unlabeled datasets [44], which allows the learner to estimate the right side of (G.1).

It is known that many instances of algorithmic bias are caused by distribution shift between the training data and real-world data encountered by the model during deployment. Broadly speaking, research has identified two types of algorithmic bias caused by distributional shifts [1]:

1. the model is trained to predict the wrong target;

2. the model is trained to predict the correct target, but its predictions are inaccurate for demographic groups that are underrepresented in the training data.

In statistical terms, the first type of algorithmic bias is caused by *posterior drift* between the training and real-world data. This leads to a mismatch between the model's predictions and the correct values of the target in the real world. The second type of algorithmic biases arises when ML models are trained or evaluated in non-diverse training data, so the models perform poorly on underserved groups. In statistical terms, this type of algorithmic bias is caused by *covariate shift* between the training and real-world data.

Several prior works study the effects of enforcing algorithmic fairness under distribution shift. Blum *et al.* [45] consider the effects of enforcing demographic parity and equalized odds under two forms of distribution shift they call under-representation bias and labeling bias. Maity *et al.* [34] consider the effects of enforcing group fairness in a domain generalization setting when there is subpopulation shift between the source and target domains. Another line of work considers how fairness guarantees (instead of performance guarantees) transfer under distribution shift [5], [46], [47]. Singh *et al.* [48] and Rezaei *et al.* [49] consider both transferability of performance and fairness guarantees under covariate shift.