# OpenReview forum: "Domain Adaptation meets Individual Fairness. And they get along."
_NeurIPS.cc/2022/Conference — NeurIPS 2022 Accept_

### Official Review · Reviewer_FXf2 · 2022-06-30

**Rating:** 7
**Confidence:** 3
**Soundness:** 4 excellent
**Presentation:** 2 fair
**Contribution:** 4 excellent

**Summary:**

This work mainly focuses on individual fairness problem under covariate shift. The authors find that individual fairness and domain adaptation methods can benefit from each other as below.

1. The regularizers for enforcing individual fairness can help models adapt to new domains. The authors consider the transduction and inductive settings. Under the standard smoothness and convexity assumptions on the regularizer and the loss function, the authors give a bound on the target-domain risk of models that are regularized by individual fairness.
2. On a synthetic setting where the source and target domain have the opposite sensitive attribute, the authors show that a domain invariant transformation appeals to individual fairness.

The authors also validate their theory on two textual datasets.

**Questions:**

1. Line 89: the ML model does not learn this function because the function is not in the model class. Why is the function not in the model class? If I am applying a deep neural networks for classification, it is likely the class of neural networks includes an approximate function that can recognize the non-western brides.
2. Line 94: the accuracy of the predictions on the unlabeled samples from the target domain in the training data. This is confusing. The unlabeled samples in the target domain is not used for training, right? Do you mean test data?
3. What is the implication of the transformation $T$ in Equation (2.9)?
4. The bound in Theorem 2.9 is independent of $epsilon$. That said, the bounds still hold for very large $epsilon$. Isn’t it vacuous?
5. What is the individual fairness metric in Table 1? Is it evaluated in the source domain or the target domain?
6. How do you interpret the condition $\Phi b = 0$ in Theorem 3.1? Is it realistic?

------
After rebuttal:
The authors have addressed my comments. I would increase my score from 6 to 7 accordingly.

**Limitations:**

I did not foresee any negative societal impact in this work.

**Strengths And Weaknesses:**

**Strengths**:
In general, the presented conclusions are intriguing and theoretically grounded. I checked the proof sketch but didn’t touch the proof details. The paper is quite clearly written. I didn’t find any problem in understanding the assumptions and the theories. The paper flow is lucid. The scope of the studied problem may have significant impacts on both the individual fairness and domain adaptation fields.

**Weaknesses**:
I would suggest the authors to significantly improve the structure of this paper. There should be a preliminary section for better understanding the problem settings as well as some key concepts. In particular, the settings of domain adaptation need to be further explained. I would also suggest the authors to complement a separate section to introduce prior works on algorithmic fairness under domain shift.

A large body of recent works on algorithmic fairness under distribution shift is missing, including:

> [1] H. Singh, R. Singh, V. Mhasawade, and R. Chunara, “Fairness violations and
mitigation under covariate shift,” in Proceedings of the 2021 ACM Conference
on Fairness, Accountability, and Transparency, 2021, pp. 3–13.
> [2] J. Schrouff, N. Harris, O. Koyejo, I. Alabdulmohsin, E. Schnider, K. OpsahlOng, A. Brown, S. Roy, D. Mincu, C. Chen et al., “Maintaining fairness across distribution shift: do we have viable solutions for real-world applications?” arXiv preprint arXiv:2202.01034, 2022.
> [3] A. Rezaei, A. Liu, O. Memarrast, and B.D. Ziebart, Robust Fairness under Covariate Shift. AAAI, 2021.
> [4] Y. Chen, R.P. Raab, J. Wang, and Y. Liu, “Fairness Transferability Subject to Bounded Distribution Shift.” arXiv preprint arXiv:2206.00129, 2022.

---

> ### Author Response · Authors · 2022-08-02
> **Response to Reviewer FXf2**
>
> Thank you for taking the time and effort to review our paper.
> We will add an additional section providing background on domain adaptation and discussing prior works on algorithmic fairness under distribution shift in the camera-ready version, as you suggested.
>
> Please find the answers to your questions below (in the same order as they appear in the original review in the **Questions** section):
>
> 1. For neural networks that are universal approximators, model misspecification is replaced by inductive bias misspecification. Model classes that include universal approximators are often overparameterized, so the learner must appeal to some inductive bias to fit a model (e.g. the implicit bias of fitting the model with SGD). Thus although the model class includes a model that recognizes non-western brides, the learner does not pick this model in practice.
>
> 2. In the *transductive setting*, the learner is given a set of labeled samples and another set of unlabeled samples. The goal is correctly predicting the labels of the given unlabeled samples; the learner is unconcerned with the accuracy of the model on new test samples. This is different from the *inductive setting*, where the goal is correctly predicting the labels of new test samples. The features of the unlabeled samples (but not their labels) are used for training in both settings. We provide theoretical results for both settings.
>
> 3. T is the optimization/decision variable in the optimization problem in (2.9) (which is (2.10) in the revised version). The optimal $T$ is an adversarial map that maps $X$ to an adversarial example $X' = T(X)$ that maximizes the difference $f(X) - g(X')$. In the domain generalization setup, when we don't have access to the unlabeled data from the target domain, we need to enforce a regularizer that provides some uniform guarantees across all domains in the vicinity of the source domain. $T$ here is used to produce the adversarial test domain example.
>
> 4. The bound in Theorem 2.9 implicitly depends on *epsilon* through $R$. For very large epsilon, $R$ is a very stringent consistency regularizer; it is large even if $f_0$ is even slightly non-smooth with respect to the fair metric. Thus the $R(f_0,f_0)$ term in the bound will be large, leading to a vacuous bound.
>
> 5. We interpret the reviewer's question in two ways:
>
> (a) *What is the fair metric used to train the IF models considered in Table 1?* The fair metric for training the IF models was obtained following the prior works proposing the corresponding IF methods. For example, in the Bios dataset, we created a variation of the original biography by flipping the gender pronouns, computed the matrix of differences in the embedding space with the original Bios embeddings, and performed SVD on this matrix to obtain ``sensitive directions'' which are then projected out when computing the fair metric (see Section 4.2 in reference [3] for additional details). In our experiments, the fair metric was obtained using only source data.
>
> (b) *What is the measure of individual fairness in Table 1?* This experiment studies the utility of IF methods for domain adaptation. Metrics in Table 1 are domain adaptation metrics, i.e. various performance metrics in the target domain.
>
> 6. The condition $\phi b = 0$ is the goal of some methods for learning fair representations. For example, in the paper *Man is to Computer Programmer as Woman is to Homemaker? Debiasing Word Embeddings*, the span of the vector $b$ is a gender subspace, and $\phi b = 0$ implies the rows of $\phi$ (embeddings of words) are gender neutral because they have no component in the gender subspace.

---

> > ### Comment · Reviewer_FXf2 · 2022-08-05
> > **Thank you for your clarification**
> >
> > Thank you for the response. It would be great if you can list the changes in the revised paper. Please see my comments below:
> >
> > 1. Thanks for the clarification. Please revise the wording in line 89. "because the function is not in the model class" is not accurate.
> > 2. I'd appreciate it if you can have a separate problem statement or preliminary section. It would be great if you can clarify the difference between _transductive_ and _inductive_ settings in a preliminary section.
> > 3. I still did not see any interpretation of the variable $T$ in the revised paper.
> > 4. Thanks for the clarification.
> > 5. (b) The individual fairness metrics should also be reported in Table 1, otherwise we don't know to what extent the compared methods boost the target domain performance while preserving the individual fairness.
> > 6. In this case, the assumption seems to be a bit strong. And the _debiasing word embeddings_ method is only considering the word embeddings of text data. How can Theorem 3.1 be generalized to tabular and image data?

---

> > > ### Author Response · Authors · 2022-08-08
> > > **Response to Reviewer FXf2**
> > >
> > > Thank you for responding! We have posted a summary of the main changes as you suggested. Please see our responses to your comments below.
> > >
> > > > "Thanks for the clarification. Please revise the wording in line 89. "because the function is not in the model class" is not accurate."
> > >
> > > We have changed line 89 of the main draft as per your suggestion.
> > >
> > > > "I'd appreciate it if you can have a separate problem statement or preliminary section. It would be great if you can clarify the difference between transductive and inductive settings in a preliminary section."
> > >
> > > We now have added a brief discussion highlighting the difference between inductive and transductive setting just before subsection 2.1 of the revised draft. We will add a broader section describing the problem setting, providing relevant domain adaptation and individual fairness background, and discussing fairness under distribution shifts in the final version of the paper. This will require some reorganization of the other sections, so we prefer to defer it to the camera-ready version to avoid major changes to the submission (and correspondingly line numbers) during the discussion period.
> > >
> > > > "I still did not see any interpretation of the variable $T$ in the revised paper."
> > >
> > > We have now added an interpretation of $T$ just after equation (2.10) in the revised draft.
> > >
> > > > "The individual fairness metrics should also be reported in Table 1, otherwise we don't know to what extent the compared methods boost the target domain performance while preserving the individual fairness."
> > >
> > > We added a new Table 3 in Appendix E (we now include the appendix as part of the main paper PDF for convenience) that compares individual fairness (measured with prediction consistency) of the methods compared in Table 1. We see that all IF methods improve individual fairness of the baseline as intended. We also refer to it in the main text in line 313.
> > >
> > > > "In this case, the assumption seems to be a bit strong. And the debiasing word embeddings method is only considering the word embeddings of text data. How can Theorem 3.1 be generalized to tabular and image data?"
> > >
> > >
> > > The result is generally applicable to data that can be modeled with a factor model. Factor models are usually used to model tabular data, but they can also be used in conjunction with representation learning to model non-tabular data, including text and images. We referred to the debiasing word embeddings paper as an example to this effect. For images, factor models can be used to model the latent codes of generative adversarial networks (e.g. see "Image Counterfactual Sensitivity Analysis for Detecting Unintended Bias" by Denton et al.).
> > >
> > > *If we have successfully addressed your questions, we would strongly appreciate an increased score. Otherwise, please let us know what experiments and/or revisions we can provide to allay your concerns.*

---

### Official Review · Reviewer_w6vp · 2022-07-07

**Rating:** 5
**Confidence:** 3
**Soundness:** 3 good
**Presentation:** 3 good
**Contribution:** 3 good

**Summary:**

This paper shows that satisfying individual fairness (IF) and representation alignment methods can complement each other. Enforcing IF is shown to mitigate algorithmic biases caused by covariate shift as long as the regression function satisfies IF. Conversely, IF can be enforced by aligning the distributions of the features under a factor model. Experiments verify the theoretical results.

**Questions:**

* Does IF improve domain adaptation when used together (and vice versa)?
* How realistic are the assumptions, and do they actually hold on the benchmark datasets?

**Limitations:**

* The authors say the limitations are stated in the paper, but it is not clear where exactly.
* There does not seem to be negative societal impact.

**Strengths And Weaknesses:**

Strengths
* Shows interesting connections between IF and domain adaptation methods.
* Provides theoretical evidence for the above connections.
* Empirical results on various datasets show that IF can be used for domain adaptation and vice versa.

Weaknesses
* Overall the compatibility between IF and domain adaptation seems to only hold in restrictive settings. IF is similar to domain adaptation only for covariate shifts, but not necessarily for other data shifts (label and concept shifts). Fundamentally, IF means similar samples get similar predictions, so it obviously cannot address all data shifts in general. Also domain adaptation is only useful to IF for specific factor model structures. I think the paper would be more interesting if IF techniques are used to improve domain adaptation technique (stated as future work), but the current contribution focuses on their similarities only. It could be that using IF along with domain adaptation does not help domain adaptation at all and vice versa, but I do not see any discussion about whether the two complement, subsume, or possibly have a negative effect on each other.
* Several assumptions are made to make the theory work without much justification, and it is not clear how realistic they are. There should be more convincing explanations and possibly empirical evidence.
  * Assumption 2.1: this seems to assume that the penalty of the regression function $f_0$ is always small ($\leq \delta$). Wouldn't this result in trivial regression and penalty functions?
  * Assumption 2.2: why should we believe that R is strongly convex?
  * Assumption 2.3: why is it reasonable that L is both strongly convex and strongly smooth?
  * Assumption 2.6: why is it reasonable to assume that R satisfies strong convexity?
* In the experiments, the assumptions are not verified on the real datasets. There are only end results for IF and domain adaptation.

======

The author response addresses my concerns, so I am increasing my score.

---

> ### Author Response · Authors · 2022-08-02
> **Response to Reviewer w6vp**
>
> We thank the reviewer for the thorough review and feedback. We address the concerns raised below.
>
> > IF is similar to domain adaptation only for covariate shifts
>
> The covariate shift assumption is standard in transfer learning, and we focus on it in the main paper because it leads to the cleanest results. That said, we recognize this as the main limitation of our results (and we have added a comment on it in the Conclusion section). To address this limitation, we also have theoretical results that relax the covariate shift assumption (covering label shift and concept drift) in Appendix E.
>
>
> > Does IF improve domain adaptation when used together (and vice versa)?
>
> In this paper, we show that *enforcing IF can be a domain adaptation method in itself* both theoretically and empirically. The methods for enforcing IF that we study can also be viewed as consistency regularization in the domain adaptation/generalization literature: the fair metric defines the neighborhoods in which consistency is enforced. Thus our results can also be understood as using fairness to improve/extend domain adaptation methods. We have added a comment on this in the conclusion.
>
>
> > Discussion of assumptions.
>
> Assumption 2.1: This assumption is not really necessary. The out-of-distribution generalization bound in Theorem 2.4 and 2.6 remain valid as stated, but they are weak or vacuous if $R(f_0)$ is large. Note a small $R(f_0)$ simply implies $f_0$ is smooth with respect to the fair metric; it does not imply $f_0$ is constant. The fair metric between two inputs that do not need to be treated similarly is large, so $f_0$ can have different outputs on these inputs without penalty.
>
>
> Assumption 2.2/2.6: These two assumptions are the transductive and inductive versions of the same assumption. In fact, we show that Assumption 2.6 implies Assumption 2.2 with high probability in Appendix D. We also check this assumption for the Laplacian regularizer in section 2.1 (at the end of Page 4, after line 159). The Laplacian regularizer is strongly convex as long as the underlying graph is connected. We also check that the analog of the Laplacian regularizer in the inductive setting is strongly convex in Appendix C. In our computational experiments, we use the Laplacian regularizer with a connected graph, so the regularizer is strongly convex.
>
>
> Assumption 2.3: The assumption that the loss function is strongly convex and strongly smooth is standard in learning theory and optimization. Note that, the quadratic loss, which is frequently used for continuous response, is strongly convex and smooth.
>
> To summarize, the assumptions we make are verifiable and easily satisfied in real data.

---

### Official Review · Reviewer_NDHi · 2022-07-12

**Rating:** 3
**Confidence:** 3
**Ethics Flag:** Yes
**Soundness:** 2 fair
**Presentation:** 2 fair
**Contribution:** 2 fair

**Summary:**

In this paper, the authors first show that under some assumptions (smoothness and convexity on the), enforcing individual fairness (IF) on the source distribution can potentially improve the performance of ML models on the target distribution, and then propose a training mechanism to add an IF regularizer into the training process to achieve individual fairness in the target distribution. The authors also show that domain adaptation algorithms that align the feature distributions in the source and target domain can be used to improve IF.

**Questions:**

Line 81, what does "the model class is mis-specified" mean? What is the function space $\mathcal{F}$ here?
More to the point, if "mis-specification" means that you can only train a simple model on the source data, say linear, but the target data is quadratic, then I disagree that this is the difficulty with domain adaptation, even if you can train an optimal quadratic model on the source, the distribution shift will likely make it to be a much worse model on the data, even though this model is in the target hypothesis class.





**Ethics Review Area:**

["Discrimination / Bias / Fairness Concerns"]

**Limitations:**

The example of wedding dress provided in section 2 feels uncomfortable, it almost seems to associate the development stage of a country with its brides wearing white dresses for weddings. You can avoid this by simply talking about a specific image dataset that contains photos of European-style wedding gowns, and note that it lacks examples of wedding dresses from a specific and contrasting culture.

**Strengths And Weaknesses:**

The idea that enforcing individual fairness can help with distribution shift tasks is a natural idea and using regularization is also an intuitive way of achieving such an objective. The paper also provides a bridge between individual fairness and domain adaption literature.


The major weakness of this paper is that it's not clear why the main results are insightful, in particular, theorem 2.4 and theorem 2.7. It seems to simply say that the target loss can be bounded as a source loss plus the regularizer's penalties, which seems natural, perhaps even obvious. I wish the bound could be more concrete and interpretable. One way of fixing this problem is to add a corollary for theorem 2.4 for specific regularizers, which will help convince the reader that this bound is non-trivial for classes of regularizers they may use.

Another problem of this paper is the meaning and the interpretation of their bounds. For example, in theorem 2I would like to see $n_s$ and $n_t$ exposed in the bound in Theorem 2.4 because those variables describe the size of the input, thus the bound would be more insightful if the authors could explain why the bound is proportional to $n_s$ and inversely proportional to $n_t$.

---

> ### Author Response · Authors · 2022-08-02
> **Response to Reviewer NDHi**
>
> We thank the reviewer for the feedback and provide answers to the raised concerns below.
>
> > Insights of the main results.
>
> We have added a paragraph (see the Example, namely *Laplacian regularizer* after line 167) providing a concrete example of an individual fairness regularizer satisfying the assumptions of Theorem 2.4. The key insight is that enforcing IF on source data improves the (bound on) target domain performance. In particular, Theorem 2.4 and 2.7 provide bounds on the risk of the fitted model in the target domain, implying individually fair models perform well in the target domain despite covariate shift between source and target domains. This is also verified in our experiments. This is surprising because the methods for enforcing IF that we study were not developed with distribution shifts in mind.
>
> We also note that Theorems 2.4 and 2.7 are out-of-distribution risk bounds (i.e. when training and test data come from different distributions), while standard learning theoretic risk bounds are in-distribution (i.e. when training and test data have the same distribution). In other words, Theorems 2.4 and 2.7 provide guarantees on the extrapolation performance of individually fair ML models.
>
> At a higher level, the main takeaway from our paper for practitioners is enforcing fairness can help out-of-distribution generalization, so even non-altruistic practitioners should consider enforcing fairness to improve the performance of their models.
>
>
> > Interpretation of $n_s$ and $n_t$ in Theorem 2.4.
>
> We revised the statement of Theorem 2.4 to make the dependence on $n_s$ and $n_t$ explicit.
> $n_s$ and $n_t$ are basically normalizing constants. They appear in Theorem 2.4 because the regularizer $\mathcal{R}$ is not properly normalized. It is possible to absorb the $n_s$’s and $n_t$’s into the other constants ($L$, $\lambda$, $\mu$ etc); this is done in the inductive version of Theorem 2.4, Theorem 2.7. We want to emphasize that the bound depends on both $n_s$ and $n_t$, but the dependence on the ratio is due to our presentation of the bound, (i.e. $\alpha_n, \beta_n$ depend on the ratio as we have already scaled the source loss and the regularizer).
>
>
> > What does “the model class is mis-specified” mean?
>
> Model misspecification means the model class $\mathcal{F}$ does not include the *true* regression function $f_0$. We would like to emphasize that our theorems include both *distribution shift* and *model misspecification*, i.e. we always consider changes in distribution (note that source and target distribution is denoted by $P$ and $Q$ respectively). To understand why the assumption of model misspecification is interesting, note that under covariate shift, if the model is well-specified, then the domain adaptation problem is easy because it is possible to learn $f_0$ consistently only from the source domain (provided sufficient data). Going back to your example, the covariate shift condition implies source and target domains share a common quadratic regression function, so a quadratic model fitted in the source domain (modulo statistical estimation error) will be the minimum MSE model in the target domain.
>
>
> > The wedding dress example
>
> This example is from the Google Inclusive Images Competition. We made no claims regarding the association between the development stage of a country and its geographic location or wedding traditions. We have reworded the example to clarify. Please let us know if this has addressed your ethics concerns.

---

> ### Author Response · Authors · 2022-08-08
> **Please check whether we addressed your questions and concerns**
>
> Dear Reviewer NDHi,
>
> Since only one day remains in the discussion period, we would appreciate it if you check and reply to our response to your comments soon. This will give us time to address further questions and comments that you may have before the end of the discussion period. Please also take a look at the summary of changes to the draft. Among them, as you suggested, we
> - Clarified the insights of our main theorems (see paragraph after line 167)
> - Revised Theorem 2.4 to make the dependence on $n_s$ and $n_t$ explicit
> - Reworded the wedding dress example and clarified model misspecification. See lines 80-91.
>
> *If our response adequately addresses your concerns, please consider raising the score of our submission. Thank you very much for your time.*

---

### Review · Ethics_Reviewer_cKaQ · 2022-08-03

**Recommendation:**

I do not think that any further revisions are strictly needed. However, I do think the paper only very briefly touches upon potential limitations and broader impacts. In my opinion, the authors could use the extra camera ready page to elaborate on this.

**Ethics Review:**

I did not see major ethical issues. There was one potential ethical issue around the wording of an example but, at least in my opinion, the authors have handled that issue adequately already.

---

### Review · Ethics_Reviewer_ZvZu · 2022-08-18

**Recommendation:**

Very simple to address, repeating text from ethics review here:
This can be addressed by making the language more clear around the traditions we are discussing, who/which communities follow them. For instance, naming countries/cultural groups and having citations in my opinion clears up the problem.

**Ethical Issues:**

Yes

**Ethics Review:**

**Very minor point** - I somewhat agree with the reviewer who flagged the "wedding dress" example. The authors are correct that they do not make an association with the economic status of a country and wedding traditions. However, I think that the reviewer correctly identifies that the language is clunky here.

This can be addressed by making the language more clear around the traditions we are discussing, who/which communities follow them. For instance, naming countries/cultural groups and having citations in my opinion clears up the problem.

---

### Author Response · Authors · 2022-08-05
**Discussion period**

We wish to thank the reviewers for helping us improve the paper. Hopefully, you had a chance to take a look at our responses. Please let us know whether we have addressed your concerns and questions. We look forward to having a fruitful discussion and would appreciate if you consider increasing the score in case our responses have addressed your concerns.

---

### Author Response · Authors · 2022-08-08
**Summary of changes**

We thank the reviewers for providing suggestions and comments that have helped us to improve the paper. We summarize the main changes in the draft below.

1. We have added a discussion on the difference between transductive and inductive setting just before subsection 2.1

2. We added an interpretation of the variable $T$ introduced in subsection 2.3 after equation (2.10). This subsection extends our theory to the domain generalization setup, where we don't have unlabeled data from the target domain.

3. We have added a paragraph after Theorem 2.4 (just after line 167) that provides a concrete example of an individual fairness regularizer satisfying the assumptions of Theorem 2.4.

4. We have revised the statement of Theorem 2.4 to make the dependence on $n_s$ and $n_t$ explicit.

5. We have now mentioned the relation between using IF regularization for domain adaptation and the consistency regularization for domain adaptation in conclusion.

6. We added a new Table 3 in Appendix E (we now include the appendix as part of the main paper PDF for convenience) that compares individual fairness (measured with prediction consistency) of the methods compared in Table 1. We see that all IF methods improve individual fairness of the baseline as intended. We also refer to it in the main text in line 313.

7. Reworded the wedding dress example and clarified model misspecification. See lines 80-91.

---

### Meta-Review · Area_Chair_h9zn · 2022-08-24

**Recommendation:** Accept
**Confidence:** Less certain

**Metareview:**

The initial reviews were divergent. During the rebuttal and discussion phase, however, many of the raised concerns are addressed propertly, leading slightly towards accept. While some of the issues are not checked yet for whether to address them, I believe the authors' response answer them adequately. Hence I recommend the acceptance of this paper.

**Award:**

No

---

### Decision · Program_Chairs · 2022-09-14

Accept